# Continual Reinforcement Learning by Planning with Online World Models

Zichen Liu [* 1 2]  Guoji Fu [* 2]  Chao Du [1]  Wee Sun Lee [2]  Min Lin [1]

## Abstract

Continual reinforcement learning (CRL) refers to a naturalistic setting where an agent needs to endlessly evolve, by trial and error, to solve multiple tasks that are presented sequentially. One of the largest obstacles to CRL is that the agent may forget how to solve previous tasks when learning a new task, known as *catastrophic forgetting*. In this paper, we propose to address this challenge by planning with online world models. Specifically, we learn a Follow-The-Leader shallow model online to capture the world dynamics, in which we plan using model predictive control to solve a set of tasks specified by any reward functions. The online world model is immune to forgetting by construction with a proven regret bound of $\mathcal{O}(\sqrt{K^2 D \log(T)})$ under mild assumptions. The planner searches actions solely based on the latest online model, thus forming a FTL *Online Agent (*OA*)* that updates incrementally. To assess OA, we further design *Continual Bench*, a dedicated environment for CRL, and compare with several strong baselines under the same model-planning algorithmic framework. The empirical results show that OA learns continuously to solve new tasks while not forgetting old skills, outperforming agents built on deep world models with various continual learning techniques.

## 1 Introduction

Continual reinforcement learning (CRL) (Khetarpal et al., 2022; Abel et al., 2024) quests for agents that can continuously evolve to solve possibly infinite number of tasks that are revealed sequentially to them. From a task-level definition, a very strong baseline is to just train a separate agent for each task, and use the information of the task ID to select the agent. Many existing works more or less fall within

this "multitask" view of CRL because task-specific components are constructed and used (*e.g.*, separate weights, task boundaries or IDs) (Kirkpatrick et al., 2017; Mallya & Lazebnik, 2018; Chaudhry et al., 2019a; Kessler et al., 2022; Gaya et al., 2023).

However, a more naturalistic scenario would involve a learning agent that seamlessly interacts and learns in an environment where there is no clear cut tasks. To distinguish with the above, we call such agent an *Online Agent* (OA). We expect the solution of OA to contain learning components that are **shared** through out the agent's lifetime, and can be **updated incrementally**.

Existing CRL methods are not building towards online agents because of two challenges. First, not all RL components have their online alternatives. For example, the value function and policy are dependent on the reward function defined by a task, thus it is innately not a learning component that can be shared across tasks. To alleviate this issue, prior works often learn them per-task (Chaudhry et al., 2019a; Wołczyk et al., 2021). Second, for components that can be shared, they should be learned incrementally without forgetting under severe data distributional shift. Huang et al. (2021) has attempted this with model-based RL, but they still rely on per-task buffer and embedding.

In this paper, we aim to develop an OA to tackle the CRL problem. We first notice that a **unified world dynamics** is the key component that can be shared across tasks and should be learned, and planning with the learned model gives us a RL agent that maximizes long-term rewards (Garcia et al., 1989; Hutter, 2000). Since the planner usually has no trainable parameters (De Boer et al., 2005; Williams et al., 2015), the goal of OA is to learn an online world model with low regret incrementally. To this end, we employ the recent Follow-The-Leader (FTL) shallow models (Liu et al., 2024) which permit efficient online updates for world modelling, and plan with the learned models using cross-entropy method (CEM) for acting. Under mild assumptions, we theoretically show the sparsely updating models (Liu et al., 2024) are no-regret, providing a definitive answer for the design of OA. Consequently, OA's every interaction with the real world brings new information to learn a more accurate world model without forgetting, which immediately benefits the planning at the next step.

---

[*]Equal contribution  [1]Sea AI Lab  [2]National University of Singapore. Correspondence to: Min Lin <linmin@sea.com>.

*Proceedings of the 42nd International Conference on Machine Learning*, Vancouver, Canada. PMLR 267, 2025. Copyright 2025 by the author(s).

To evaluate OA in the CRL settings, we further develop Continual Bench, which explicitly focuses on the design of a unified world dynamics. It also takes account of forgetting and transfer simultaneously and is computationally lightweight (see motivations in Section 2 part 2). The empirical results demonstrate the superior performance of OA over strong CRL baselines under the same model-planning framework, suggesting its great promise in developing future autonomous artificially agents.

## 2 Related Work

In this section, we provide an overview of how our work integrates with existing literature. Since this work tries to contribute in both algorithm and benchmark aspects, the discussion is structured into the following two subsections.

**Algorithm perspective**. The most related work to ours is Liu et al. (2024), where the authors propose a non-linear encoding technique and an analytic update rule to learn world models online. We follow their work to employ similar online shallow networks to learn the world dynamics. To bridge the gap between practice and theory, we further analyze the convergence of the online sparse model learning, and identify that regularization is necessary to guarantee it converges to the offline optimal solution. Our work is also methodology-wise distinct from Liu et al. (2024), which is based on the Dyna architecture (Sutton, 1990) and still learns model-free components such as value and policy functions using model-synthetic data. This could further compound difficulties of applying model-free methods to develop continual agents, such as the need to maintain a replay buffer (Isele & Cosgun, 2018), the loss of plasticity (Abbas et al., 2023), and even the dependency on task-specific models (Garcia & Thomas, 2019; Wołczyk et al., 2021). In the supervised CL settings, (Zhuang et al., 2022; 2023; 2024) also employ an analytic solution to solve class incremental learning without forgetting (an FTL approach), but they rely on pretrained deep features over all tasks, which is impossible for online agents. On the contrary, we propose to use a model-planning framework to tackle the CRL problem, given a unified world dynamics representation and external reward function, making our agent (OA) potential to generalize to new tasks zero-shot (Sancaktar et al., 2022).

**Benchmark perspective**. CRL has typically been evaluated on a sequence of Atari games (Kirkpatrick et al., 2017; Schwarz et al., 2018; Rolnick et al., 2019). CORA (Powers et al., 2022) recently extends this conventional protocol to a set of image-based discrete control environments, additionally including procedurally-generated environments (Cobbe et al., 2020; Küttler et al., 2020) and realistic home simulators (Kolve et al., 2017; Shridhar et al., 2020). Their focus on image-based environments stems

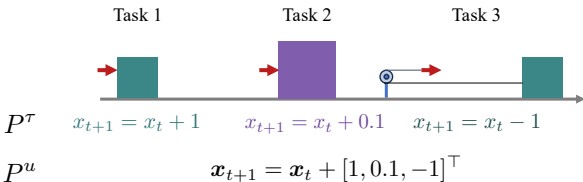

$$P^\tau \qquad x_{t+1} = x_t + 1 \qquad x_{t+1} = x_t + 0.1 \qquad x_{t+1} = x_t - 1$$

$$P^u \qquad \boldsymbol{x}_{t+1} = \boldsymbol{x}_t + [1, 0.1, -1]^\top$$

**Figure 1:** A motivating example comparing task-indexed dynamics ($P^\tau$) with unified dynamics ($P^u$). Under the task-indexed viewpoint, the state space of each task is the box's location $x \in \mathbb{R}$ while the action space is {"apply force rightwards", "apply force leftwards"}. Clearly, for the same action "apply force rightwards" (denoted as red arrows) applied at the same state $x_t$, the dynamics $P^\tau$ is conflicting to each other. Hence, there is **no solution** for a CRL agent to model the dynamics, unless incorporating task IDs. In contrast, the unified dynamics $P^u$ treats each task individually and builds a "global" view of the world, where the solutions to all tasks can co-exist without the need of task IDs.

from the need for a consistent shared observation space, which they choose to be the pixel space. The games included are also visually-distinct with little to no overlapping, making the benchmark suitable for studying *forgetting*. However, the image-based environments demand prohibitive computation resources, and the lack of meaningful overlapping prevents us from evaluating the transfer of CRL algorithms. Continual-World (Wołczyk et al., 2021), on the other hand, focuses on continuous control and *transfer*, using a sequence of 10 robot manipulation tasks selected from Meta-World (Yu et al., 2019). Nevertheless, their state-based inputs encode the positions of different objects at fixed locations, which results in *inconsistent state space* (further explained in Section 5), hindering the original purpose of studying transfer. In this work, we propose a lightweight but realistic benchmark environment, Continual Bench, that has a consistent state space and takes both forgetting and transfer into consideration.

## 3 Preliminaries

### 3.1 Continual reinforcement learning

We first review some necessary RL basics and then define the setting of continual RL. RL is usually formulated as a Markov decision process (MDP) (Bellman, 1957) $\mathcal{M} = (\mathcal{S}, \mathcal{A}, P, R, \gamma, \rho_0)$, where $\boldsymbol{s}_t \in \mathcal{S}$ and $\boldsymbol{a}_t \in \mathcal{A}$ are state and action at time step $t$, $P : \mathcal{S} \times \mathcal{A} \mapsto \mathcal{S}$ is the transition dynamics, $R : \mathcal{S} \times \mathcal{A} \mapsto \mathbb{R}$ is a reward function associated with a specific task, $\gamma \in (0, 1)$ is a discount factor, and $\rho_0$ is the initial state distribution. An RL agent aims to *identify an optimal policy* $\pi : \mathcal{S} \mapsto \mathcal{A}$ that maximizes long-term reward (or return): $\pi^* = \arg\max_\pi \mathbb{E}_{\pi, P} \left[ \sum_{t=0}^\infty \gamma^t R(\boldsymbol{s}_t, \boldsymbol{a}_t, \boldsymbol{s}_{t+1}) \mid \boldsymbol{s}_0 = \boldsymbol{s} \sim \rho_0 \right]$. A continual RL agent (Abel et al., 2024), on the other hand, needs to *conduct indefinite search* for policies to control a sequence of MDPs $\mathcal{M}_\tau = (\mathcal{S}, \mathcal{A}, P^u, R^\tau, \gamma, \rho_0^\tau)_\tau$ with a consistent and shared state and action space. We also consider a unified dynamics $P^u$ that describes the whole

world an agent can see, instead of time-indexed pieces $P^\tau$ (Wołczyk et al., 2021; Khetarpal et al., 2022) that can only be distinguished by explicit task IDs, or otherwise becoming POMDPs whose task ID information needs to be inferred (Nagabandi et al., 2018). We depict a motivating example in Figure 1 to give intuitions on the differences between $P^\tau$ and $P^u$. While our definition offers a more simplified formulation, it does not compromise nonstationarity, the core problem in CRL, because the nonstationary reward function $R^\tau$ will drive the CRL agent to perform different tasks, incurring distributional shift in state-action visitation. This gives the objective of a continual agent

$$J_T(\pi) := \frac{1}{T}\sum_{\tau=1}^{T}\mathbb{E}_{\pi,P^u}\left[\sum_{t=0}^{\infty}\gamma^t R^\tau(\boldsymbol{s}_t, \boldsymbol{a}_t, \boldsymbol{s}_{t+1})|\boldsymbol{s}_0 = \boldsymbol{s} \sim \rho_0^\tau\right].$$

(1)

Note that $T$ in Eq. (1) is the number of tasks the agent has seen till now, instead of the total number of tasks, which is an unknown for a continual agent. In other words, a continual agent is expected to gain high return for all experienced tasks. In this paper, we develop such an agent by learning a world model online, and form its policy by planning with the learned model.

### 3.2 Planning and model predictive control

**Planning**. Given the world dynamics $P$ and reward function $R$, planning involves optimizing action sequences $\boldsymbol{a}_{t:t+H}$ to maximize the $H$-step finite-horizon return given current state $\boldsymbol{s}_t$:

$$\boldsymbol{a}_{t:t+H}^\star = \arg\max_{\boldsymbol{a}_{t:t+H}} \sum_{i=0}^{H} R\left(\boldsymbol{s}_{t+i}, \boldsymbol{a}_{t+i}\right)|\boldsymbol{s}_{t+1} \sim P(\boldsymbol{s}_t, \boldsymbol{a}_t).$$

(2)

As a model-based RL method, planning has the advantage that it can reuse the same world dynamics and optimize for different reward functions without adaptation. This is especially of our interest in developing a continual agent, because when facing a new task the agent can directly utilize prior knowledge about the world to attempt to solve it.

**Model predictive control** (MPC) synthesizes a closed-loop policy from the planning results. It takes the first action $\boldsymbol{a}_t$ from the optimal action sequence, executes it in the environment and re-plans based on the new environment state. An MPC-based agent collects experiences following such policy, while improving its internal estimation about the world through experiences (Negenborn et al., 2005) for better planning. In this paper, we focus on the unified world dynamics learning driven by an external time-varying reward function, which aligns with our CRL formulation.

## 4 Online Agent for Continual Reinforcement Learning

In this section we present the proposed Online Agent (OA). Section 4.1 introduces the online world model learning pro-

cess, followed by our results on the regret of the sparse model learning (Section 4.2). Based on the learned no-regret model, Section 4.3 describes how the policy is constructed using planning.

### 4.1 Online world model learning

We present the learning process of OA, where we employ shallow but wide networks that support efficient Follow-The-Leader (FTL) online learning to model the world. In particular, we learn a model of the form $\boldsymbol{y} = \boldsymbol{W}\sigma(\boldsymbol{P}\boldsymbol{x})$, where $\boldsymbol{P}$ is a projection matrix filled with fixed random values drawn from a Gaussian distribution, $\sigma$ is an activation function and $\boldsymbol{W}$ contains learnable weights. Such architecture allows universal approximation (Huang et al., 2006) but requires a very high dimensional hidden layer to have great capacity. Following Liu et al. (2024), we learn a linear model online per time step, using a high-dimensional sparse feature encoder for $\sigma(\boldsymbol{P}\boldsymbol{x})$ and solving for $\boldsymbol{W}$ in closed form with FTL strategy.

**Modeling**. Let $S$ and $A$ denote the dimenionality of state and action spaces. When the agent is in state $\boldsymbol{s}_t$ it executes an action $\boldsymbol{a}_t$ and observes how the world changes. The observation may incur a loss thus correcting its previous perception of the world dynamics, which is modeled linearly by $\boldsymbol{y}_t = \phi(\boldsymbol{x}_t)^\top\boldsymbol{W}$. $\boldsymbol{W} \in \mathbb{R}^{D\times S}$ is the linear layer, $\boldsymbol{x}_t = [\boldsymbol{s}_t, \boldsymbol{a}_t] \in \mathbb{R}^{S+A}$ is the input for the world model and $\phi(\boldsymbol{x}_t) \in \mathbb{R}^D$ is its sparse high-dimensional projection, and $\boldsymbol{y}_t = (\boldsymbol{s}_{t+1} - \boldsymbol{s}_t) \in \mathbb{R}^S$ the state differences. At the start of time step $t$, we have $\boldsymbol{\Phi}_{t-1} = [\phi(\boldsymbol{x}_1), \ldots, \phi(\boldsymbol{x}_{t-1})]^\top \in \mathbb{R}^{(t-1)\times D}$ accumulating all the inputs and $\boldsymbol{Y}_{t-1} \in \mathbb{R}^{(t-1)\times S}$ as well for targets. A FTL world model can be solved by regularized least squares:

$$\forall t, \boldsymbol{W}^{(t)} = \arg\min_{\boldsymbol{W}\in\mathbb{R}^{D\times S}} \|\boldsymbol{\Phi}_{t-1}\boldsymbol{W} - \boldsymbol{Y}_{t-1}\|_F^2 + \frac{1}{\lambda}\|\boldsymbol{W}\|_F^2,$$

(3)

where $\|\cdot\|_F$ denotes the Frobenius norm.

**Learning** the world model simply requires finding a solution to Eq. (3), which can be obtained analytically as $\boldsymbol{W}^{(t)} = (\boldsymbol{\Phi}_{t-1}^\top\boldsymbol{\Phi}_{t-1} + \frac{1}{\lambda}\boldsymbol{I})^{-1}\boldsymbol{\Phi}_{t-1}^\top\boldsymbol{Y}_{t-1}$. Though this computation does not grow with the amount of data, the inversion could still be inefficient since $D$ is usually large for modeling complex environments. We thus utilize the feature sparsity and conduct incremental update on the model:

$$\boldsymbol{W}_s^{(t)} = \left(\boldsymbol{A}_{ss}^{(t-1)} + \frac{1}{\lambda}\boldsymbol{I}\right)^{-1}(\boldsymbol{B}_s^{(t-1)} - \boldsymbol{A}_{s\bar{s}}^{(t-1)}\boldsymbol{W}_{\bar{s}}^{(t-1)}),$$

(4)

where $\boldsymbol{A}^{(t-1)} = \boldsymbol{\Phi}_{t-1}^\top\boldsymbol{\Phi}_{t-1}$ and $\boldsymbol{B}^{(t-1)} = \boldsymbol{\Phi}_{t-1}^\top\boldsymbol{Y}_{t-1}$, $s$ contains all the indices on which the latest input feature $\phi(\boldsymbol{x}_{t-1})$ has non-zero values, *i.e.*, the newly coming data point only activates $K = |s| \ll D$ nodes, and $\bar{s}$ is its complement. Using them in subscript means taking a sub-matrix from the original one. The activation ratio $K/D$ is

fixed and small for any input to guarantee a constant and low update overhead. Note that when $s$ covers all coordinates (thus the feature is dense), Eq. (4) recovers the analytic solution to Eq. (3). We refer to Appendix A.1 for a more detailed sparse feature construction process.

Despite the resemblance to the update rule in Algorithm 2 of Liu et al. (2024), Eq. (4) highlights a crucial difference that $\frac{1}{\lambda}\boldsymbol{I}$ derived from the ridge regularization term is important for the local update to have a unique minimizer (Lemma 1), or otherwise it may not converge (Peng & Vidal, 2023) when $\boldsymbol{A}_{ss}^{(t-1)}$ is rank-deficient, which is the case during the initial phase of the learning when data points are only a few. We provide theoretical analysis in Section 4.2.

## 4.2 Regret analysis

In this section we show the sparse update rule (Eq. (4)) learns a no-regret ($\mathcal{O}(\log(T))$) world model. With the following mild assumptions for hyperparameters and inputs, we have Theorem 1.

**Assumption 1.** *For all $t \geq 1$ and $\boldsymbol{x}_1, \boldsymbol{x}_2, \ldots, \boldsymbol{x}_t$, we assume that there exists $\lambda \geq 1$ such that*

$$\sup_{\boldsymbol{x}}\left\|\phi(\boldsymbol{x})\phi(\boldsymbol{x})^\top - \frac{1}{t}\sum_{i=1}^{t}\phi(\boldsymbol{x}_i)\phi(\boldsymbol{x}_i)^\top\right\|_2 \leq \frac{1}{\lambda t}. \quad (5)$$

**Assumption 2.** *Assume that for all $t \geq 1$, $\|\boldsymbol{y}_t\|_2 \leq c_y$ with $y_{t,i} \geq 0$, and $\|\boldsymbol{W}\|_F \leq c_W$.*

**Assumption 3.** *For all $t \geq 1$, we set $K$ such that*

$$K \geq \min\Big\{D,$$

$$c_y^2\Big(\frac{\sum_{i=1}^{t}\phi(\boldsymbol{x}_i)\phi(\boldsymbol{x}_i)^\top}{\phi(\boldsymbol{x}_t)^\top\phi(\boldsymbol{x}_t)}+1\Big)\sqrt{\left\|\boldsymbol{A}_{\bar{s}s}^{(t)}\big(\boldsymbol{A}_{ss}^{(t)}+\frac{1}{\lambda}\boldsymbol{I}\big)^{-1}\right\|_2^2+1}\Big\}.$$

**Remark 1** (Discussions on the assumptions). *Assumption 1 ensures that as more data points are observed, the feature mappings stabilize. The difference between the outer product of any new input's feature mapping and the average outer product of the observed mappings decreases as increases. In essence, as more data is gathered, the empirical covariance matrix better approximates the true covariance matrix. The term $\frac{1}{\lambda t}$ controls this stabilization rate, with $\lambda$ as a scaling factor. It ensures the model's representations become more reliable over time, improving generalization to new data. In practice, Assumption 1 holds if we explore new points near to the observed data, i.e., within $\frac{1}{\lambda t}$ away from the center in the feature space. Assumption 2 assumes bounded input and output, which is a mild condition. Assumption 3 states that the choice of the hyperparameter $K$ depends on two factors: (1) the weights of the features at time $t$ over all observed data and (2) the sparsity of the activated elements in the feature matrix $\boldsymbol{A}^{(t)}$. Intuitively, If $\boldsymbol{A}_{ss}$ is more significant than $\boldsymbol{A}_{\bar{s}s}$*

*in spectral norm, or if the feature value at $\boldsymbol{x}_t$ is larger than previously observed data, a smaller $K$ suffices to ensure model performance. In practice, we choose $K$ such that Assumption 3 hold.*

**Theorem 1** (Regret Bounds for Sparse Online Model Learning). *Let $\widetilde{\boldsymbol{W}}^{(1)}, \widetilde{\boldsymbol{W}}^{(2)}, \ldots$ be the sequence produced by the sparse online model learning Eq. (4). Let $r_M \geq 1$ be a constant defined as in Eq. (18). Suppose that there exist $c_W, c_y > 0$ and $\lambda \geq 1$ such that Assumptions 1 to 3 hold. Then, for all $T \geq 1$ and all $\xi \in \mathbb{R}^{D \times S}$, we have*

$$\text{Regret}(T) := \sum_{t=1}^{T} f_t(\widetilde{\boldsymbol{W}}^{(t)}) - \sum_{t=1}^{T} f_t(\xi)$$

$$\leq \frac{1}{\lambda}c_W^2 + 5\lambda(K^2 r_M^2 + 1)c_y^2 D(\log(T) + 1). \quad (6)$$

*If $\lambda = c_W/c_y\sqrt{5(K^2 r_M^2 + 1)D(\log(T) + 1)} \geq 1$ and it satisfies Assumptions 1 and 3, then we have*

$$\text{Regret}(T) \leq c_W c_y\sqrt{20(K^2 r_M^2 + 1)D(\log(T) + 1)}. \quad (7)$$

*Proof Sketch (for formal proof see Appendix E.3.2).* Note that we can decompose the regret into:

$$\text{Regret}(T) = \sum_{t=1}^{T}\Big(f_t(\widetilde{\boldsymbol{W}}^{(t)}) - f_t(\boldsymbol{W}^{(t)})\Big)$$

$$+ \Big(\sum_{t=1}^{T} f_t(\boldsymbol{W}^{(t)}) - \sum_{t=1}^{T} f_t(\xi)\Big).$$

The first term involves the gap between the solutions of Eq. (3) and the sparse online model learning Eq. (4) at each time step. By Schur's Complement Lemma for block matrix inversion (Zhang, 2006) together with the sparse update rule Eq. (4), we can upper-bound the difference between $\widetilde{\boldsymbol{W}}^{(t)}$ and $\boldsymbol{W}^{(t)}$ (see Proposition 3), which is further used to bound $f_t(\widetilde{\boldsymbol{W}}^{(t)}) - f_t(\boldsymbol{W}^{(t)})$ (see the proof in Appendix E.3.2). The second term is the regret bound for online model learning Eq. (3). By bounding the difference between $\boldsymbol{W}^{(t)}$ and $\boldsymbol{W}^{(t+1)}$ (see Lemma 2), we can upper-bound $f(\boldsymbol{W}^{(t)}) - f(\boldsymbol{W}^{(t+1)})$ (see Lemma 3). Then, following a similar proof (Shalev-Shwartz et al., 2012) for the regret bounds of FTL models, we obtain the regret bound for online model learning (see Corollary 1). Combining the bounds for the two terms we conclude the proof. We refer to Appendices E.2 and E.3 for more details and other results. □

## 4.3 Planning with online world models

Based on its internal no-regret model about the world, OA acts in the environment by planning and MPC. For planning we use the cross-entropy method (CEM) (Rubinstein,

1999; De Boer et al., 2005), a stochastic derivative-free optimization technique that has demonstrated effectiveness in different model-based RL scenarios (Chua et al., 2018; Wang & Ba, 2020). It solves Eq. (2) in an iterative manner as described below.

In each iteration, it first generates $N$ action sequence candidates $\mathcal{C} = \{\boldsymbol{a}_{t:t+H}\}_{n=1}^N$ for a planning horizon $H$, in which each $\boldsymbol{a}_i$ is independently sampled from $\mathcal{N}(\boldsymbol{\mu}, \text{diag}(\boldsymbol{\sigma}^2))$ with $\boldsymbol{\mu}, \boldsymbol{\sigma} \in \mathbb{R}^A$. Then, individual candidates are evaluated by simulating the action sequences in the learned model to compute the total rewards. Finally, only a fixed number of elite candidates with high total rewards is selected and used to estimate the parameters $\{\boldsymbol{\mu}, \boldsymbol{\sigma}\}$ by maximum likelihood for the next iteration. After a few iterations, an approximately optimal action sequence can be found and used by MPC.

Instead of the plain CEM, we adopt several improvements to make it efficient for model-based RL. The first modification is *shift-initialization*. At time step $(t+1)$, we initialize the candidates $\mathcal{C}_{t+1}^0$ using the solution of the previous decision step $\mathcal{C}_t^\star$, shifted by one step. This is because MPC only takes the first best action and discards the remaining $(H-1)$ results, which could be utilized to provide better initialization for the next time step. We also incorporate *colored noise* and *memory* proposed by Pinneri et al. (2021) for better sample efficiency. The colored noise injects temporal correlation along the planning horizon when generating $\boldsymbol{a}_{t:t+H}$, which potentially leads to deeper exploration of the state space. Meanwhile, the memory keeps the elite candidates generated at each CEM iteration and carry a fraction of them over to initialize the next iteration. Reusing these high quality samples could speed up the convergence of CEM iterations. We refer to Appendices A.2 and A.3 for a detailed algorithm and hyperparameters used in this paper.

## 5 Continual Bench: An Environment for CRL Evaluation

The field of machine learning in general benefits from properly designed datasets and benchmarks to iterate algorithms. However, in CRL, there are not many widely adopted benchmarks, and prior works tend to themselves create a sequence of tasks from an existing suite of environments. For example, Kirkpatrick et al. (2017) select a sequence of Atari games (Bellemare et al., 2013) to learn sequentially, and Huang et al. (2021) vary the world properties (*e.g.* the density of the same object) in a single environment. The former type may overlook knowledge transfer across tasks due to the *lack of meaningful overlapping*, and the latter kind could be *unrealistic* because real world physics does not change.

Recently, Wołczyk et al. (2021) propose a benchmark

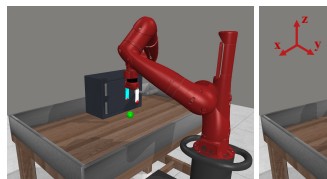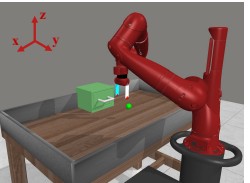

**Figure 2:** Potential physical conflicts between two consecutive tasks in Continual-World (Wołczyk et al., 2021). The handles of the door and the drawer follow different trajectory when being opened.

named Continual-World, where they select a sequence of tasks from Meta-World (Yu et al., 2019) and run experiments using soft actor-critic (SAC) (Haarnoja et al., 2018) with various continual learning techniques. This benchmark has put explicit focus on low-level transfer, because the same robotic arm is instructed to achieve different goals by interacting with real-world objects. However, the potential underlying *physical conflict* impedes the original design purpose.

We use Figure 2 to illustrate how such conflict would appear and why it is undesirable. The figure shows two similar tasks: open the door (left) and open the drawer (right). As different tasks all share the same observation space (Wołczyk et al., 2021), the locations of both the door handle and the drawer handle will be regarded as "the first object location", thus their values are placed into the same positions of the observation vector. However, when the gripper pulls the handle, the door will allow a circular movement in the $xy$ plane, but the drawer will constrain the displacement to the $y$ axis, immediately resulting in conflicting world dynamics. This could prevent us from studying forgetting or transfer in CRL, because there is even no common solution about the world, not to mention how the skills learned from one task can transfer to the next one. This issue is not obvious in Wołczyk et al. (2021) because they work with model-free methods, take advantage of task ID information, and learn separate heads for different tasks.

Motivated by above-mentioned issues, we design Continual Bench, a dedicated environment for CRL evaluation, for assessing OA and comparing it with baselines. Our development is based on the task primitives proposed by Meta-World, but instead of directly concatenating several tasks temporally like Continual-World, we redesign the environment to arrange different tasks *spatially*. This ensures the existence of a *unified world dynamics*, which corresponds to the multi-task solution, a basic assumption for achieving an ideal continual learner (Peng et al., 2023). Figure 3 gives an overview of the environment. Starting from the lower left and anti-clockwise, the 6 selected tasks with diverse difficulty levels are: peg-unplug, faucet-close, pick-place, door-open, window-close, button-press. They are placed in a circle with maximal distances to each other, facilitating test on forgetting in the presence of dis-

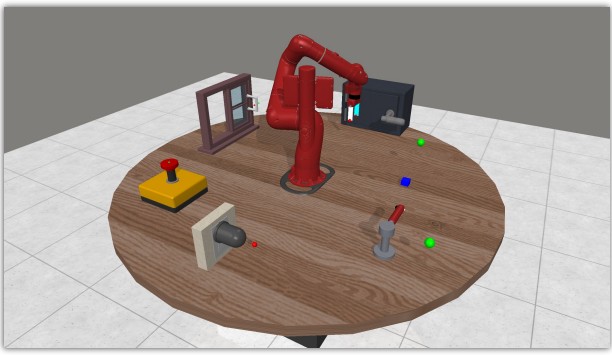

**Figure 3:** The Continual Bench environment consists of 6 tasks with diverse difficulty levels. All tasks share the same unified dynamics.

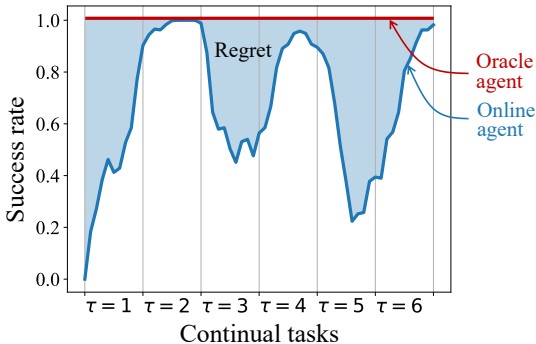

**Figure 4:** Regret can be calculated with the area in blue, which integrates the sub-optimality gap of the online agent.

tasks to evaluate agents at global time step $w$:

$$AP(w) = \frac{1}{T_w} \sum_{\tau=1}^{T_w} p_\tau(w), \quad \text{with}$$

$$p_\tau(w) = \mathbb{E}_{\pi, P^u} \left[ \prod_{t=0}^{\infty} \mathbb{I}(\|s_t - g\|_2^2 < \delta) | s_0 = s \sim \rho_0^\tau \right]. \quad (8)$$

$T_w$ is the number of tasks an agent has experienced at step $w$, $p_\tau(w)$ is the success rate of the policy at time $w$ on task $\tau$, and $\mathbb{I}$ is the indicator function. Intuitively, this metric reflects the *offline* performance of $\pi$ at time step $w$ across all observed tasks, thus accounting for forgetting.

We also define **regret** to better capture the *online* performance of an agent:

$$Reg(w) = \frac{1}{w} \int_{\tau=0}^{w} (1 - p_{T_\tau}(\tau)) d\tau, \quad (9)$$

which effectively calculates the normalized area enclosed by the online agent's performance curve and that of an oracle agent (with success rate always being 1), as depicted in Figure 4.

tributional shift in state-action visitation when switching tasks. Different tasks also share meaningful overlapping, allowing us to study transfer, though it is not the focus of this work. We open source the code of Continual Bench[1] and hope this realistic but lightweight environment can accelerate the progress of CRL research. Please see Appendix B for detailed environment specifications.

## 6 Experiment

We evaluate `OA` using Continual Bench, and compare its CRL capability with several continual learning baselines under the same agent design framework.

### 6.1 Setup and metrics

We mainly focus on the model-based planning framework, where all agents we compare are model-based RL agents that learn a world model using experiences and act using CEM planning with MPC (refer to Figure 7). In this setup, `OA` learns the world model online (Section 4.1), while the baselines employ a deep world models, a default choice of many prior approaches (Chua et al., 2018; Wang & Ba, 2020; Kessler et al., 2023).

The reward function is provided to the agent and changed upon task switch, but no task boundary information is given to the world model learning (unless the continual learning baseline requires it). We let all agents run in Continual Bench to continuously solve a sequence of 6 tasks in the order (`pick-place`, `button-press`, `door-open`, `peg-unplug`, `window-close`, `faucet-close`). The order is chosen such that adjacent tasks are as spatially far away as possible to increase the nonstationarity.

The performance is measured in accordance to the CRL objective defined in Eq. (1). Since all the tasks in Continual Bench have a binary success measure depending on current and goal state, we define **average performance** on all seen

### 6.2 Baselines

Continual learning (CL) methods can be categorized into several paradigms, such as regularization-based methods (Kirkpatrick et al., 2017; Zenke et al., 2017; Aljundi et al., 2019b), replay-based methods (Riemer et al., 2018; Aljundi et al., 2019a; Chaudhry et al., 2019b) and architecture-based methods (Mallya & Lazebnik, 2018; Kang et al., 2022). Prior CRL works typically apply CL methods to base RL agents, such as the model-free Soft Actor-Critic (SAC) (Haarnoja et al., 2018; Yang et al., 2023) or the model-based Dyna (Sutton, 1990; Liu et al., 2024). We follow the practices to test multiple representative CL methods with both a model-free actor-critic agent and a model-based planning agent. Concretely, we consider **EWC** (Kirkpatrick et al., 2017) and **SI** (Zenke et al., 2017) for regularization-based methods, **PackNet** (Mallya

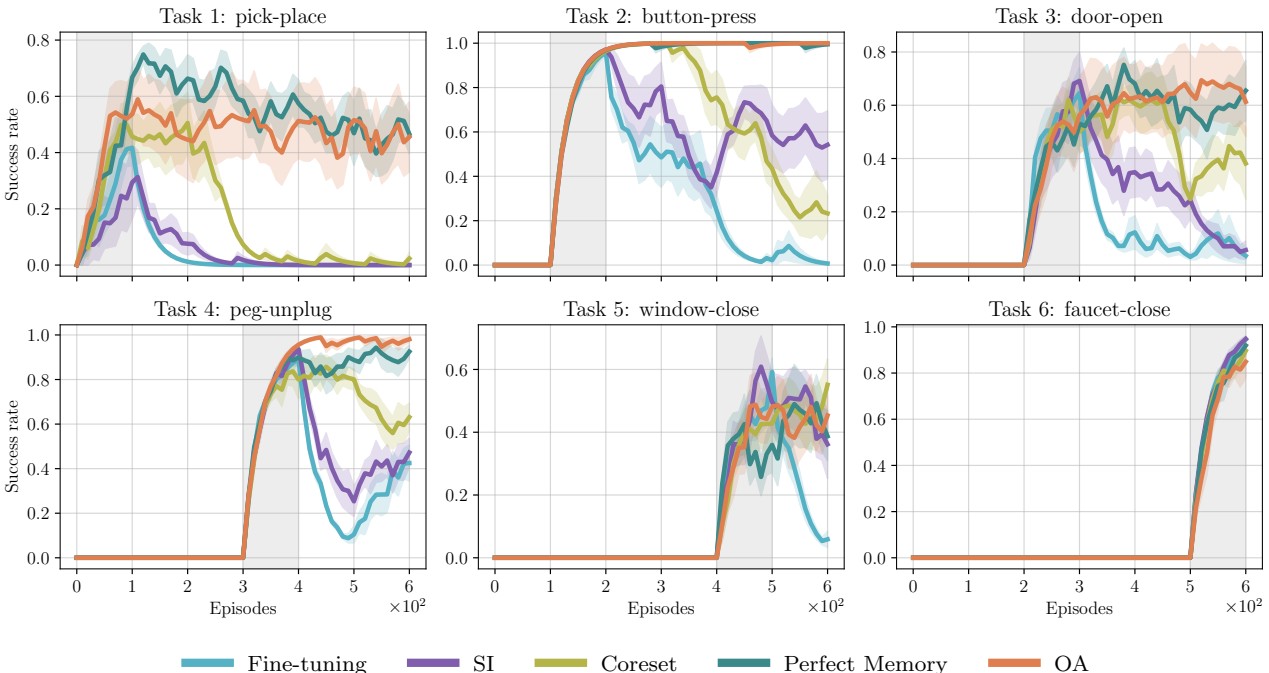

**Figure 5:** Performance comparison of our agent `OA` with different deep model-based planning agents on Continual Bench environment. Results are aggregated from 7 runs with different seeds. The solid lines show the means and shaded areas are the standard errors. The grey regions denote the learning period for the current task $\tau \in [1, 2, \ldots, 6]$.

& Lazebnik, 2018) for architecture-based methods, and **Coreset** (Vitter, 1985) for replay-based methods. We also include the vanilla baseline of **Fine-tuning** which fine-tunes the model only on the current task, and upper-bound baselines **Perfect Memory** that train on all previous data. More details about the baselines can be found in Appendix A.5.

### 6.3 Learning curves of model-based agents

We first present the comparison of learning curves between `OA` and baseline agents in the model-based planning setting on the Continual Bench environment. Figure 5 shows the performance (success rate) of different agents for all 6 tasks, measured along the learning process. Different tasks are presented to the agents sequentially, with the shaded region in grey denoting the learning period of that particular task. We first observe that during the learning period of a particular task, all agents effectively learn to perform the task well, showing the *plasticity* of model-based agents: it can reuse the knowledge of the world dynamics and quickly adapt to new tasks by planning with corresponding reward functions. Focusing on the *stability* (to retain the performance on previously seen tasks), we notice that **Fine-tuning** agents immediately cease to perform well on the old task when switching to the new one due to catastrophic forgetting. Using **SI** to regularize the weight updates can alleviate the forgetting slightly, resulting in a bit higher final performance on two tasks (`button-press` and

`window-close`). Maintaining a **Coreset** for experience replay performs better than **SI**, resembling the state-of-the-art performance of replay-based methods in supervised continual learning (Boschini et al., 2022). However, its performance diminishes as the proportion of older experiences decreases. This suggests that determining the size of the replay is a challenging design choice, especially when the number of tasks and the number of samples in each task are unknown. In comparison, `OA` stands out by demonstrating non-forgetting capability building on the online FTL world models. For all tasks it has seen, `OA` maintains a high performance over all remaining time steps, matching the performance of **Perfect Memory**. However, **Perfect Memory** ensures non-forgetting by keeping all the data and performing SGD updates until convergence, which is inefficient with unboundedly growing computation for lifelong agents. `OA`, on the other hand, achieves the same performance by a much more efficient online update with constant overhead.

### 6.4 Average performance of model-based and model-free agents

We show the average performance (Eq. (8)) curves over all seen tasks in Figure 6(a). We scale the performance by $\frac{T_w}{6}$ for better visual effect. The dashed lines denote task switching, upon which we can observe a performance jump of different scales for all agents. Due to the loss of stability, deep agents (even with various continual learning tech-

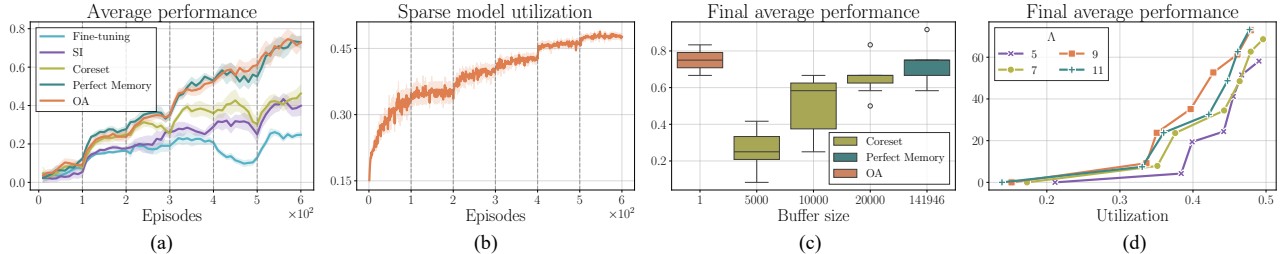

**Figure 6: (a)** The average performance curves of different methods. **(b)** Ratio of the activated weights of the OA sparse world model. **(c)** Final agent performance with different buffer budgets. **(d)** Ablation on the world model sparsity.

**Table 1:** Comparison on average performance and regret (in %) of regularization-based (🏴), architecture-based (🔧), replay-based (📚) and perfect memory (⛓) methods.

| Base | | Methods | $AP$ (↑) | $Reg$ (↓) |
|---|---|---|---|---|
| Model-free (SAC) | | Fine-tuning | 0.69 | 67.68 |
| | 🏴 | EWC | 31.45 | 77.38 |
| | 🔧 | PackNet | 41.72 | 78.50 |
| | 📚 | Coreset | 37.31 | 77.97 |
| | ⛓ | Perfect Memory | 41.51 | 77.58 |
| Model based | | Fine-tuning | 24.86 | 37.74 |
| | 🏴 | SI | 39.96 | 33.57 |
| | 📚 | Coreset | 61.83 | 30.83 |
| | ⛓ | Perfect Memory | **73.09** | 30.95 |
| | | OA (ours) | **72.93** | **27.62** |

niques) struggle to succeed on all sequentially seen tasks. Only OA (and deep agent with perfect memory) can improve the average performance continually. Table 1 gives quantitative results of the average performance (AP) and the regret (Reg) measured over all tasks at the final step of agent learning. For model-free agents, various CL methods improve AP compared to the Fine-tuning baseline, while they all incur high regret. This might be because the learned policy and value are hard to adapt to new tasks quickly. On the other hand, model-based agents generally exhibit lower regret by planning with the learned world model, which is shared across tasks thus being more generalizable. Notably, the proposed method achieves even lower regret than deep agents with perfect memory. This is because our incremental update ensures the optimal solution at each step, while SGD updates on all previous data are not guaranteed and could be less efficient. While model-based results in Figure 5 & Table 1 measure the agent's CRL performance (a joint result of world model and planner), the world model accuracy provides a more direct measure. Please see Appendix C for more results.

### 6.5 Ablation analysis

OA perceives the state-action inputs as high-dimensional sparse features (Section 4.1), and naturally learns a sparse

world model. Figure 6(b) shows the change of the model utilization (ratio of activated weights) of OA. Interestingly, we can observe the sparse model gradually grows its utilization as learning more new tasks. Note that even if OA reaches nearly full model utilization, it can still learn from newly coming data because the model solves for an overall least squares solution based on sufficient statistics (Eq. (3)) and the sparse learning is no-regret (Theorem 1). In contrast, methods based on iterative solution finding from a prior solution subspace (Farajtabar et al., 2020; Peng et al., 2023) may exhaust the solution space quickly and fail to learn from new data. Figure 6(c) compares the performance against the buffer size, and shows that OA achieves the best performance under buffer constraints. Finally, in Figure 6(d) we ablate the effect of different sparsity of the model. $\Lambda$ denotes the number of bins in Losse ((Liu et al., 2024), Appendix A.1); larger $\Lambda$ means sparser models. The results show that sparser model contains greater capacity and reaches higher performance with less activated weights. This is a natural result as $D$ increases with $\Lambda$ (the network is becoming wider), but the attractive property is that both the update and inference computation remains almost the same since $K$ is fixed. This shows the flexibility of choosing suitable sparsity levels to model environments with different complexities.

## 7 Conclusion

In this paper, we propose to tackle the CRL problem by developing Online Agents, which should learn a shared component throughout the agent's lifetime and update incrementally. In environments with a unified world dynamics across tasks, we learn such a shared component by online FTL world modeling and act by planning. Theoretically we prove that the sparse online update learns a no-regret world model. To assess the agent's CRL capability, we further develop a benchmark named Continual Bench. Empirical results show that our Online Agents outperform several strong baselines under a fair setting, demonstrating its effectiveness and promise in building future autonomous agents.

## Impact Statement

This paper presents work whose goal is to advance the field of Machine Learning. There are many potential societal consequences of our work, none which we feel must be specifically highlighted here.

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

# A  Agent details

In this section we provide more details about OA, including the sparse non-linear feature encoding, the learning and acting process, the model hyperparameters, and the planing algorithm. We also include baseline and experimental details for reproducibility.

## A.1  Sparse feature encoding

We revisit the feature construction process of Losse (Liu et al., 2024), on which we will build our online world models for planning. At time step $t$, given an input vector $\boldsymbol{x}_t \in \mathbb{R}^{S+A}$, we first randomly project it to obtain bounded random features $\sigma(\boldsymbol{x}_t) = \text{sigmoid}(\boldsymbol{P}\boldsymbol{x}_t)$, where $\boldsymbol{P} \in \mathbb{R}^{d \times (S+A)}$ is sampled from a multivariate Gaussian $\mathcal{N}(\boldsymbol{0}, \frac{1}{S+A}\boldsymbol{I})$ to preserve feature similarity (Johnson & Lindenstrauss, 1984), and sigmoid is applied element-wise to ensure each feature value is bounded between $(0, 1)$, such that the binning edges can be clearly defined. Next, each element of $\sigma(\boldsymbol{x}_t)$ is binned softly by locating its neighboring edges and computing the distances from them. We illustrate using 1-dimensional feature grid with $\Lambda$ bins. For the $i$-th element of $\sigma(\boldsymbol{x}_t)$, let $I_i = (\Lambda - 1) \cdot \sigma(\boldsymbol{x}_t)_i$ denote its projection location on the grid. Then the "activated" indices are

$$s_i = [\lfloor I_i \rfloor, \lfloor I_i \rfloor + 1],$$

and their associated values are

$$v_i = [1 - (I_i - \lfloor I_i \rfloor), I_i - \lfloor I_i \rfloor].$$

Therefore, the $i$-th slice of the resulting feature vector is a $\Lambda$-long zero vector with non-zero values filled at indices $s_i$ with values $v_i$, and the final feature $\phi(\boldsymbol{x}_t)$ is the concatenation of $d$ such slices.

## A.2  The learning and acting process

> **Online Agent (OA) Learning and Acting Loop**
>
> **Require:** zero-initialized agent memories $\boldsymbol{A}^{(0)}$, $\boldsymbol{B}^{(0)}$ and world model weights $\boldsymbol{W}^{(0)}$; sparse encoder $\phi : \mathbb{R}^{S+A} \mapsto \mathbb{R}^D$; initial index $s = [D]$; planner CEM; sequence of tasks $(R^\tau)_{\tau=1}^\infty$; initial state $\boldsymbol{s}_1 \sim \rho_0$; time step $t = 1$.
>
> 1: **loop**                                                                    ▷ OA runs forever, updates per step
> 2:   **if** task changes **then**
> 3:     $\boldsymbol{\mu}_t \leftarrow$ init values $\in \mathbb{R}^{A \times H}$
> 4:   **else**
> 5:     $\boldsymbol{\mu}_t \leftarrow$ shifted $\boldsymbol{\mu}_{t-1}$ (fill the last column with init values)
> 6:   **end if**
>
> 7:   $\boldsymbol{W}_s^{(t)} \leftarrow (\boldsymbol{A}_{ss}^{(t-1)} + \frac{1}{\lambda}\boldsymbol{I})^{-1}(\boldsymbol{B}_s^{(t-1)} - \boldsymbol{A}_{s\bar{s}}^{(t-1)}\boldsymbol{W}_{\bar{s}}^{(t-1)})$
> 8:   $\boldsymbol{a}_t, \boldsymbol{\mu}_{t+1} \leftarrow \text{CEM}(\boldsymbol{s}_t, \boldsymbol{W}^{(t)}, \boldsymbol{\mu}_t, R^\tau)$        ▷ Appendix A.4 for details
> 9:   $\boldsymbol{s}_{t+1} \leftarrow \text{environment}(\boldsymbol{s}_t, \boldsymbol{a}_t)$
>
> 10:   $\boldsymbol{x}_t \leftarrow [\boldsymbol{s}_t, \boldsymbol{a}_t]; \boldsymbol{y}_t \leftarrow \boldsymbol{s}_{t+1} - \boldsymbol{s}_t$
> 11:   $s \leftarrow \text{nonzero\_index}(\phi(\mathbf{x}_t))$
> 12:   $\boldsymbol{A}_{ss}^{(t)} \leftarrow \boldsymbol{A}_{ss}^{(t-1)} + \phi_s(\boldsymbol{x}_t)\phi_s(\boldsymbol{x}_t)^\top$
> 13:   $\boldsymbol{B}_s^{(t)} \leftarrow \boldsymbol{B}_s^{(t-1)} + \phi_s(\boldsymbol{x}_t)\boldsymbol{y}_t^\top$
> 14:   $t \leftarrow t + 1$
> 15: **end loop**

**Figure 7:** OA learning and acting loop.

### A.3 `OA` hyperparameters

To build the sparse world model, we use 300 2-dimensional Losse features with $\Lambda = 9$. We perform a coarse sweeping over different sparsity levels ($\Lambda = (5, 7, 9, 11)$) and find $\Lambda = 5$ slightly under-fits the environment while all others give robust good performance. See Figure 6(d) for a comparison. We find $\frac{1}{\lambda} = 0.005$ as a good regularization strength without any tuning.

For the planner, we use a candidate sampling size of $N = 150$ with planning horizon $H = 15$ and $K = 3$ iterations. The ratio of elite candidates is $0.1$.

### A.4 CEM planning

We give a detailed algorithm of CEM that we use to extract policy from a learned model below.

---

**Require:** candidate sampling size $N$; number of iteration $K$; planning horizon $H$; init noise $\sigma_{\text{init}}$.

1: **function** CEM($s, W, \mu, R$)
2:     $\sigma \leftarrow$ init values $\in \mathbb{R}^{A \times H}$
3:     **for** $k \leftarrow 1, K$ **do**
        /*   *add colored noise*   */
4:         candidates $\mathcal{C} = \{a_{0:H}\}_{n=1}^{N} \leftarrow N$ samples from $\texttt{clip}(\mu + C^{\beta}(A, H) \odot \sigma)$
        /*   *memory*   */
5:         add a fraction of `elite-set` into $\mathcal{C}$ if $k > 1$
6:         $r \leftarrow$ unroll $N$ trajectories to get returns

$$\sum_{t=0}^{H} R(s_t, a_t) \mid s_0 = s, s_{t+1} = s_t + \phi([s_t, a_t])^{\top} W$$

7:         `elite-set` $\leftarrow$ best $K$ candidates
8:         $\mu, \sigma \leftarrow$ fit Gaussian distribution to `elite-set`
9:     **end for**
10: **end function**

---

### A.5 Baseline details

**Fine-tuning**. This serves as a minimal baseline for agents that use deep world models. The agent is allowed to keep all the data within the current task, but cannot carry the data over to the next task. The world model is repeatedly fine-tuned using a sequence of datasets.

**Coreset**. This refers to replay-based CL methods where a small buffer is kept across all tasks. We use reservoir sampling (Vitter, 1985), which updates the buffer such that its distribution approximates the empirical distribution of all observed samples. We use a large buffer ($10K$) in our experiments.

**Synaptic Intelligence** (Zenke et al., 2017). As a regularization-based method, synaptic intelligence (SI) estimates the importance of each parameter based on the gradients and weights deviation, and regularizes the parameter updates based on the recorded importance. Similar to **Fine-tuning** only the data of the current task can be kept, and task boundary information is needed for SI to recompute the importance measure. The two hyper-parameters used in SI, $c = 0.5$ and $\xi = 0.05$, are selected using a coarse grid search.

**Perfect Memory**. It is impractical to assume unlimited buffer capacity, but this baseline serves as the upper bound for agents using deep world models. It can also be regarded as a deep version of the FTL world model (Liu et al., 2024) – the current model is optimized using all previous data – but in an inefficient way.

In all above baselines, we use a 4-layer MLP with hidden size 200 and `ReLU` activation for world modeling. We use the Adam optimizer (Kingma & Ba, 2015) with learning rate of $4 \times 10^{-4}$ and minibatch size of 256. Since updating deep models per time step is computationally impractical, especially for CRL settings where the dataset size grows along inter-

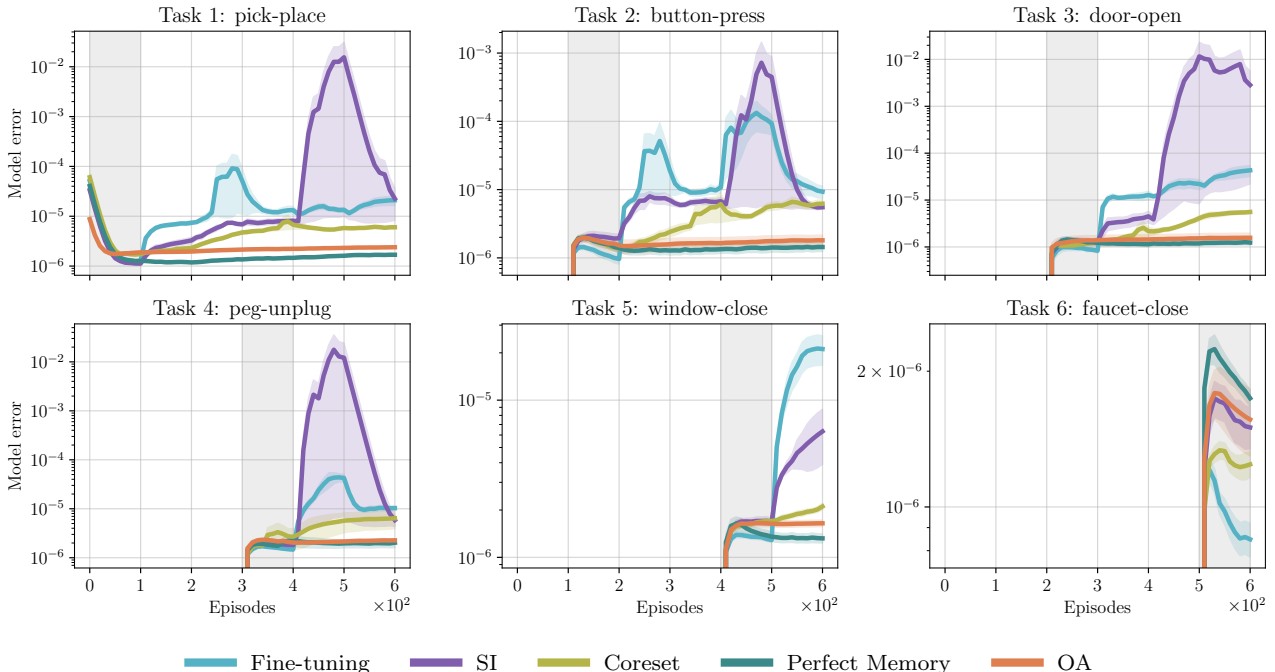

**Figure 8:** Curves of the world model errors. Same as the performance measure in the main text, results are aggregated from 7 runs with different seeds. The solid lines show the means and shaded areas are the standard errors. The grey regions denote the learning period for the current task $\tau \in [1, 2, \ldots, 6]$.

actions, we update the models every 250 environment step. On every update, the deep models are trained until convergence over all available data (stopped when the validation loss on 5% holdout data does not improve over 5 consecutive epochs). All hyperparameters of the planner are the same as OA's for a fair comparison (see Appendix A.3).

### A.6 Experimental details

All experiments in the paper are run on the internal cluster, with each job consuming one A100 GPU and 16 CPUs. The training time of a single experiment ranges from 10 hours to 15 hours, given a total of 600 episodes budget. We develop our agent and the baselines using the MBRL Library[2] (Pineda et al., 2021) by Meta (MIT license, v0.2.0).

## B Continual Bench details

We build Continual Bench based on the Mujoco (Todorov et al., 2012) physics engine and task primitives from Meta-World (Yu et al., 2019). We include in total 6 tasks (pick-place, button-press, door-open, peg-unplug, window-close, faucet-close) with diverse difficulty levels. Incorporating all the information about the world on the table, we have $\dim(\mathcal{S}) = 26$. The state includes: hand position (3), gripper openness (1), button position (3), door handle position (3), door opening angle (1), window handle position (3), faucet handle position (3), peg position (3), block position (3), gripper velocity (1), gripper pad offsets (2). The action space is the same as Meta-World, a 2-tuple consisting the change of end-effector position in $3D$ space (3) and a normalized torque applied to gripper fingers (1). When the environment starts, each episode can be at most 500-step long, with early termination on task success. The reward function $R^\tau$ is changed according to the task on hand upon switching. All reward functions are defined based on their well-crafted subroutines (Section E of Yu et al. (2019)).

## C More empirical results

We present more empirical results in this section. In Figure 8, we show the mean squared error (MSE) of world models when the agents learn sequentially on the Continual World environment, to more directly investigate the forgetting issues. We keep a separate buffer for each task, and measure the MSE of the model on the data of every seen task. We can observe that the baseline agents (except the one with perfect memory) all exhibit increasing model errors when leaving the current task, but OA can maintain nearly the same accuracy along the whole learning process.

---

[2]https://github.com/facebookresearch/mbrl-lib

## D  Limitations and Future Work

One limitation of the current OA is that the online world model can only deal with moderate-dimensional state-based observation and does not capture world uncertainties. We envisage that developing highly capable probabilistic models that permits efficient online learning is a future research avenue. Besides, though the current framework is simple yet effective, the model planing does not incorporate explicit exploration. We leave this topic for future research. The proposed benchmark environment, Continual Bench, is limited to episodic settings with explicit task switch across episodes. This is mainly due to the challenge brought by irreversible states (Sharma et al., 2022) since the underlying MDP is non-ergodic (*e.g.*, a block falling outside the table can never be placed back). Developing a reset-free CRL environment consisting of many tasks is a potential future work.

## E  Theoretical analysis and proofs

### E.1  Closed-form solutions for sparse online world model learning

**Lemma 1** (Sparse Online Model Learning Minimizer). *For every $t \in [1, T], s \subseteq [D]$, the following problem at time step $t$ has a unique global minimizer*

$$\widetilde{W}_s^{(t)} := \underset{W_s \in \mathbb{R}^{K \times S}}{\arg\min} \|\Phi_{t-1}([W_{\overline{s}}^{(t-1)}; W_s]) - Y_{t-1}\|_F^2 + \frac{1}{\lambda}\|[W_{\overline{s}}^{(t-1)}; W_s]\|_F^2. \tag{10}$$

*The global minimizer $\widetilde{W}_s^{(t)}$ has a closed-form solution as Eq. (4), i.e.:*

$$\widetilde{W}_s^{(t)} = \left(A_{ss}^{(t)} + \frac{1}{\lambda}I\right)^{-1}(B_s^{(t)} - A_{s\overline{s}}^{(t)}W_{\overline{s}}^{(t-1)}),$$

*where $\widetilde{W}^{(t)} = [W_{\overline{s}}^{(t-1)}; \widetilde{W}_s^{(t)}], A^{(t)} = \Phi_{t-1}^\top \Phi_{t-1}, B^{(t)} = \Phi_{t-1}^\top Y_{t-1}$ and they can be decomposed into*

$$A^{(t)} = \begin{pmatrix} A_{\overline{s}\overline{s}}^{(t)} & A_{\overline{s}s}^{(t)} \\ A_{s\overline{s}}^{(t)} & A_{ss}^{(t)} \end{pmatrix}, B^{(t)} = \begin{pmatrix} B_{\overline{s}}^{(t)} \\ B_s^{(t)} \end{pmatrix}, \widetilde{W}^{(t)} = \begin{pmatrix} W_{\overline{s}}^{(t-1)} \\ \widetilde{W}_s^{(t)} \end{pmatrix}.$$

*Proof.* **Uniqueness.** By the convexity of Frobenius norm and both two terms in Eq. (10) are quadratic in $W_s$, the objective function Eq. (10) is strictly convex and has a unique solution.

**Closed-form solution.** Note that

$$\|\Phi W - Y\|_F^2 + \frac{1}{\lambda}\|W\|_F^2$$

$$= \|\Phi W\|_F^2 + \|Y\|_F^2 - 2\langle \Phi W, Y \rangle_F + \frac{1}{\lambda}\|W\|_F^2$$

$$= \text{Tr}\left(W^\top \Phi^\top \Phi W\right) + \text{Tr}\left(Y^\top Y\right) - 2\text{Tr}\left(W^\top \Phi^\top Y\right) + \frac{1}{\lambda}\|W\|_F^2$$

$$= \text{Tr}\left(W^\top A W\right) + \text{Tr}\left(Y^\top Y\right) - 2\text{Tr}\left(W^\top B\right) + \frac{1}{\lambda}\|W\|_F^2$$

$$= \text{Tr}\left(\begin{pmatrix} W_{\overline{s}} \\ W_s \end{pmatrix}^\top \begin{pmatrix} A_{\overline{s}\overline{s}} & A_{\overline{s}s} \\ A_{s\overline{s}} & A_{ss} \end{pmatrix}\begin{pmatrix} W_{\overline{s}} \\ W_s \end{pmatrix}\right) + \text{Tr}\left(Y^\top Y\right) - 2\text{Tr}\left(\begin{pmatrix} W_{\overline{s}} \\ W_s \end{pmatrix}^\top \begin{pmatrix} B_{\overline{s}} \\ B_s \end{pmatrix}\right)$$

$$\quad + \frac{1}{\lambda}\text{Tr}\left(\begin{pmatrix} W_{\overline{s}} \\ W_s \end{pmatrix}^\top \begin{pmatrix} W_{\overline{s}} \\ W_s \end{pmatrix}\right)$$

$$= \text{Tr}\left(W_{\overline{s}}^\top A_{\overline{s}\overline{s}} W_{\overline{s}}\right) + \text{Tr}\left(W_s^\top A_{ss} W_s\right) + 2\text{Tr}\left(W_s^\top A_{s\overline{s}} W_{\overline{s}}\right) + \text{Tr}\left(Y^\top Y\right)$$

$$\quad - 2\text{Tr}\left(W_{\overline{s}}^\top B_{\overline{s}}\right) - 2\text{Tr}\left(W_s^\top B_s\right) + \frac{1}{\lambda}\text{Tr}\left(W_{\overline{s}}^\top W_{\overline{s}}\right) + \frac{1}{\lambda}\text{Tr}\left(W_s^\top W_s\right)$$

As a result,

$$\min_{\boldsymbol{W}_s \in \mathbb{R}^{K \times S}} \|\Phi_t(\boldsymbol{W}_{\overline{s}}; \boldsymbol{W}_s) - \boldsymbol{Y}_t\|_F^2 + \frac{1}{\lambda}\|\boldsymbol{W}\|_F$$

$$\iff \min_{\boldsymbol{W}_s \in \mathbb{R}^{K \times S}} \text{Tr}\left(\boldsymbol{W}_s^\top \boldsymbol{A}_{ss} \boldsymbol{W}_s\right) + 2\,\text{Tr}\left(\boldsymbol{W}_s^\top \boldsymbol{A}_{s\overline{s}} \boldsymbol{W}_{\overline{s}}\right) - 2\,\text{Tr}\left(\boldsymbol{W}_s^\top \boldsymbol{B}_s\right) + \frac{1}{\lambda}\text{Tr}\left(\boldsymbol{W}_s^\top \boldsymbol{W}_s\right) \tag{11}$$

Differentiate the objective function Eq. (11) w.r.t $\boldsymbol{W}_s$ and set it to zero, we obtain

$$\widetilde{\boldsymbol{W}}_s = \left(\boldsymbol{A}_{ss} + \frac{1}{\lambda}\boldsymbol{I}\right)^{-1}(\boldsymbol{B}_s - \boldsymbol{A}_{s\overline{s}}\boldsymbol{W}_{\overline{s}}).$$

$\square$

## E.2 Theoretical analysis for online model Learning

Before we derive the regret bounds for sparse online model learning Eq. (10), we first derive the regret bound for online model learning Eq. (3). The developed regret bound of online model learning, as given in Corollary 1, will be further used to derive the regret bounds for sparse online model learning Eq. (10) in Appendix E.3.2.

### E.2.1 ONLINE MODEL LEARNING

For $t \geq 1$, $i = 1, \ldots, t$, we define

$$f_i(\boldsymbol{W}) = \text{Tr}(\boldsymbol{W}^\top \phi(\boldsymbol{x}_i)\phi(\boldsymbol{x}_i)^\top \boldsymbol{W}) + \text{Tr}(\boldsymbol{y}_i^\top \boldsymbol{y}_i) - 2\,\text{Tr}(\boldsymbol{W}^\top \phi(\boldsymbol{x}_i)\boldsymbol{y}_i^\top),$$

$$\ell_{\text{reg}}(\boldsymbol{W}) = \frac{1}{\lambda}\|\boldsymbol{W}\|_F^2.$$

Then, the regularized least squares problem Eq. (3) can be rewritten as

$$\forall t, \quad \boldsymbol{W}^{(t)} = \arg\min_{\boldsymbol{W} \in \mathbb{R}^{D \times S}} \sum_{i=1}^{t-1} f_i(\boldsymbol{W}) + R(\boldsymbol{W}). \tag{12}$$

**Lemma 2.** *Let $\boldsymbol{W}^{(1)}, \boldsymbol{W}^{(2)}, \ldots$ be the sequence produced by Eq. (12) (or equivalently Eq. (3)). Suppose that there exist $\lambda \geq 1$ such that Assumption 1 holds, then for all $t \geq 1$, we have*

$$\left(1 - \frac{1}{t}\right)\boldsymbol{W}^{(t)} + \Delta_t \preceq \boldsymbol{W}^{(t+1)} \preceq \boldsymbol{W}^{(t)} + \Delta_t,$$

*where*

$$\Delta_t := \left(\sum_{i=1}^t \phi(\boldsymbol{x}_i)\phi(\boldsymbol{x}_i)^\top + \frac{1}{\lambda}\boldsymbol{I}\right)^{-1}\phi(\boldsymbol{x}_t)\boldsymbol{y}_t^\top. \tag{13}$$

*Proof.*

$$\forall t, \quad \frac{\mathrm{d}f_t(\boldsymbol{W})}{\mathrm{d}\boldsymbol{W}} = 2\phi(\boldsymbol{x}_t)\phi(\boldsymbol{x}_t)^\top \boldsymbol{W} - 2\phi(\boldsymbol{x}_t)\boldsymbol{y}_t^\top.$$

Then we have

$$\frac{\mathrm{d}\sum_{i=1}^t f_i(\boldsymbol{W})}{\mathrm{d}\boldsymbol{W}} + \frac{\mathrm{d}\ell_{\text{reg}}(\boldsymbol{W})}{\mathrm{d}\boldsymbol{W}} = \sum_{i=1}^t \frac{\mathrm{d}f_i(\boldsymbol{W})}{\mathrm{d}\boldsymbol{W}} + \frac{2}{\lambda}\boldsymbol{W} = \sum_{i=1}^t 2\phi(\boldsymbol{x}_i)\phi(\boldsymbol{x}_i)^\top \boldsymbol{W} - \sum_{i=1}^t 2\phi(\boldsymbol{x}_i)\boldsymbol{y}_i^\top + \frac{2}{\lambda}\boldsymbol{W}.$$

Therefore, we get

$$
\begin{aligned}
\boldsymbol{W}^{(t+1)} &= \left( \sum_{i=1}^{t} \phi(\boldsymbol{x}_i)\phi(\boldsymbol{x}_i)^\top + \frac{1}{\lambda}\boldsymbol{I} \right)^{-1} \left( \sum_{i=1}^{t} \phi(\boldsymbol{x}_i)\boldsymbol{y}_i^\top \right) \\
&= \left( \sum_{i=1}^{t} \phi(\boldsymbol{x}_i)\phi(\boldsymbol{x}_i)^\top + \frac{1}{\lambda}\boldsymbol{I} \right)^{-1} \left( \sum_{i=1}^{t-1} \phi(\boldsymbol{x}_i)\boldsymbol{y}_i^\top \right) + \left( \sum_{i=1}^{t} \phi(\boldsymbol{x}_i)\phi(\boldsymbol{x}_i)^\top + \frac{1}{\lambda}\boldsymbol{I} \right)^{-1} \phi(\boldsymbol{x}_t)\boldsymbol{y}_t^\top \\
&= \left( \sum_{i=1}^{t} \phi(\boldsymbol{x}_i)\phi(\boldsymbol{x}_i)^\top + \frac{1}{\lambda}\boldsymbol{I} \right)^{-1} \left( \sum_{i=1}^{t-1} \phi(\boldsymbol{x}_i)\phi(\boldsymbol{x}_i)^\top + \frac{1}{\lambda}\boldsymbol{I} \right) \boldsymbol{W}^{(t)} + \left( \sum_{i=1}^{t} \phi(\boldsymbol{x}_i)\phi(\boldsymbol{x}_i)^\top + \frac{1}{\lambda}\boldsymbol{I} \right)^{-1} \phi(\boldsymbol{x}_t)\boldsymbol{y}_t^\top \\
&= \left( \boldsymbol{I} - \left( \sum_{i=1}^{t} \phi(\boldsymbol{x}_i)\phi(\boldsymbol{x}_i)^\top + \frac{1}{\lambda}\boldsymbol{I} \right)^{-1} \phi(\boldsymbol{x}_t)\phi(\boldsymbol{x}_t)^\top \right) \boldsymbol{W}^{(t)} + \Delta_t \\
&\succeq \left( \boldsymbol{I} - \frac{1}{t}\boldsymbol{I} \right) \boldsymbol{W}^{(t)} + \Delta_t \qquad\qquad\qquad\qquad\qquad\qquad\qquad\text{(Assumption 1)} \\
&= \left( 1 - \frac{1}{t} \right) \boldsymbol{W}^{(t)} + \Delta_t .
\end{aligned}
$$

On the other hand,

$$
\begin{aligned}
\boldsymbol{W}^{(t+1)} &= \left( \sum_{i=1}^{t} \phi(\boldsymbol{x}_i)\phi(\boldsymbol{x}_i)^\top + \frac{1}{\lambda}\boldsymbol{I} \right)^{-1} \left( \sum_{i=1}^{t} \phi(\boldsymbol{x}_i)\boldsymbol{y}_i^\top \right) \\
&= \left( \sum_{i=1}^{t} \phi(\boldsymbol{x}_i)\phi(\boldsymbol{x}_i)^\top + \frac{1}{\lambda}\boldsymbol{I} \right)^{-1} \left( \sum_{i=1}^{t-1} \phi(\boldsymbol{x}_i)\boldsymbol{y}_i^\top \right) + \left( \sum_{i=1}^{t} \phi(\boldsymbol{x}_i)\phi(\boldsymbol{x}_i)^\top + \frac{1}{\lambda}\boldsymbol{I} \right)^{-1} \phi(\boldsymbol{x}_t)\boldsymbol{y}_t^\top \\
&\preceq \left( \sum_{i=1}^{t-1} \phi(\boldsymbol{x}_i)\phi(\boldsymbol{x}_i)^\top + \frac{1}{\lambda}\boldsymbol{I} \right)^{-1} \left( \sum_{i=1}^{t-1} \phi(\boldsymbol{x}_i)\boldsymbol{y}_i^\top \right) + \left( \sum_{i=1}^{t} \phi(\boldsymbol{x}_i)\phi(\boldsymbol{x}_i)^\top + \frac{1}{\lambda}\boldsymbol{I} \right)^{-1} \phi(\boldsymbol{x}_t)\boldsymbol{y}_t^\top \\
&= \boldsymbol{W}^{(t)} + \Delta_t .
\end{aligned}
$$

$\square$

**Lemma 3.** *Let $\boldsymbol{W}^{(1)}, \boldsymbol{W}^{(2)}, \dots$ be the sequence produced by Eq. (12) (or equivalently Eq. (3)). Suppose that there exist $\lambda \geq 1, c_y > 0$ such that Assumptions 1 and 2 hold. Then for all $t \geq 1$, we have*

$$
|f_t(\boldsymbol{W}^{(t)}) - f_t(\boldsymbol{W}^{(t+1)})| \leq 5\lambda c_y^2 D \frac{1}{t} .
$$

*Proof.* Recall from Lemma 2 that $\Delta_t - \frac{1}{t}\boldsymbol{W}^{(t)} \preceq \boldsymbol{W}^{t+1} - \boldsymbol{W}^{(t)} \preceq \Delta_t$. Then we have

$$
|f_t(\boldsymbol{W}^{(t+1)}) - f_t(\boldsymbol{W}^{(t)})|
$$

$$
= \left| \mathrm{Tr}(\phi(\boldsymbol{x}_t)\phi(\boldsymbol{x}_t)^\top \boldsymbol{W}^{(t+1)}\boldsymbol{W}^{(t+1)\top}) - 2\,\mathrm{Tr}(\phi(\boldsymbol{x}_t)\boldsymbol{y}_t^\top \boldsymbol{W}^{(t+1)\top}) \right.
$$

$$
\left. - \mathrm{Tr}(\phi(\boldsymbol{x}_t)\phi(\boldsymbol{x}_t)^\top \boldsymbol{W}^{(t)}\boldsymbol{W}^{(t)\top}) + 2\,\mathrm{Tr}(\phi(\boldsymbol{x}_t)\boldsymbol{y}_t^\top \boldsymbol{W}^{(t)\top}) \right|
$$

$$
= \left| \mathrm{Tr}\Big(\phi(\boldsymbol{x}_t)\phi(\boldsymbol{x}_t)^\top (\boldsymbol{W}^{(t+1)} - \boldsymbol{W}^{(t)})(\boldsymbol{W}^{(t+1)} - \boldsymbol{W}^{(t)})^\top\Big) + 2\,\mathrm{Tr}\Big(\phi(\boldsymbol{x}_t)\phi(\boldsymbol{x}_t)^\top (\boldsymbol{W}^{(t+1)} - \boldsymbol{W}^{(t)})\boldsymbol{W}^{(t)\top}\Big) \right.
$$

$$
\left. - 2\,\mathrm{Tr}(\phi(\boldsymbol{x}_t)\boldsymbol{y}_t^\top \boldsymbol{W}^{(t+1)\top}) + 2\,\mathrm{Tr}(\phi(\boldsymbol{x}_t)\boldsymbol{y}_t^\top \boldsymbol{W}^{(t)\top}) \right|
$$

$$
\leq \left| \mathrm{Tr}\Big(\phi(\boldsymbol{x}_t)\phi(\boldsymbol{x}_t)^\top (\boldsymbol{W}^{(t+1)} - \boldsymbol{W}^{(t)})(\boldsymbol{W}^{(t+1)} - \boldsymbol{W}^{(t)})^\top\Big) \right| + 2\left| \mathrm{Tr}\Big(\phi(\boldsymbol{x}_t)\phi(\boldsymbol{x}_t)^\top (\boldsymbol{W}^{(t+1)} - \boldsymbol{W}^{(t)})\boldsymbol{W}^{(t)\top}\Big) \right|
$$

$$
+ 2\left| \mathrm{Tr}\big(\phi(\boldsymbol{x}_t)\boldsymbol{y}_t^\top (\boldsymbol{W}^{(t+1)} - \boldsymbol{W}^{(t)})^\top\big) \right|
$$

$$
\leq \left| \mathrm{Tr}\big(\phi(\boldsymbol{x}_t)\phi(\boldsymbol{x}_t)^\top \Delta_t\Delta_t^\top\big) \right| + 2\left| \mathrm{Tr}\big(\phi(\boldsymbol{x}_t)\phi(\boldsymbol{x}_t)^\top \Delta_t \boldsymbol{W}^{(t)\top}\big) \right| + 2\left| \mathrm{Tr}\big(\phi(\boldsymbol{x}_t)\boldsymbol{y}_t^\top \Delta_t^\top\big) \right| \qquad \text{(Lemma 2)}
$$

$$
= \left| \mathrm{Tr}\big(\phi(\boldsymbol{x}_t)\phi(\boldsymbol{x}_t)^\top \Delta_t\Delta_t^\top\big) \right| + 2\left| \mathrm{Tr}\Big(\phi(\boldsymbol{x}_t)\phi(\boldsymbol{x}_t)^\top \underbrace{\Big(\sum_{i=1}^t \phi(\boldsymbol{x}_i)\phi(\boldsymbol{x}_i)^\top + \frac{1}{\lambda}\boldsymbol{I}\Big)^{-1}}_{\preceq \frac{1}{t}\boldsymbol{I} \text{ by Assumption 1}} \phi(\boldsymbol{x}_t)\boldsymbol{y}_t^\top \boldsymbol{W}^{(t)\top}\Big) \right|
$$

$$
+ 2\left| \mathrm{Tr}\Big(\phi(\boldsymbol{x}_t)\boldsymbol{y}_t^\top \boldsymbol{y}_t\phi(\boldsymbol{x}_t)^\top \Big(\Big(\sum_{i=1}^t \phi(\boldsymbol{x}_i)\phi(\boldsymbol{x}_i)^\top + \frac{1}{\lambda}\boldsymbol{I}\Big)^{-1}\Big)^\top\Big) \right|
$$

$$
\leq \mathrm{Tr}\big(\phi(\boldsymbol{x}_t)\phi(\boldsymbol{x}_t)^\top \Delta_t\Delta_t^\top\big) + \frac{2}{t}\left| \mathrm{Tr}\Big(\phi(\boldsymbol{x}_t)\boldsymbol{y}_t^\top \boldsymbol{W}^{(t)\top}\Big) \right| + 2C_y^2\left| \mathrm{Tr}\Big(\phi(\boldsymbol{x}_t)\phi(\boldsymbol{x}_t)^\top \underbrace{\Big(\sum_{i=1}^t \phi(\boldsymbol{x}_i)\phi(\boldsymbol{x}_i)^\top + \frac{1}{\lambda}\boldsymbol{I}\Big)^{-1}}_{\preceq \frac{1}{t}\boldsymbol{I} \text{ by Assumption 1}}\Big) \right|
$$

$$
\leq \mathrm{Tr}\big(\phi(\boldsymbol{x}_t)\phi(\boldsymbol{x}_t)^\top \Delta_t\Delta_t^\top\big) + \frac{2}{t}\left| \mathrm{Tr}\Big(\phi(\boldsymbol{x}_t)\boldsymbol{y}_t^\top \boldsymbol{W}^{(t)\top}\Big) \right| + C_y^2 D\frac{2}{t}.
$$

For the first term on the right hand side of the above inequality, we have

$$
\mathrm{Tr}\big(\phi(\boldsymbol{x}_t)\phi(\boldsymbol{x}_t)^\top \Delta_t\Delta_t^\top\big)
$$

$$
= \mathrm{Tr}\left(\phi(\boldsymbol{x}_t)\phi(\boldsymbol{x}_t)^\top \Big(\sum_{i=1}^t \phi(\boldsymbol{x}_i)\phi(\boldsymbol{x}_i)^\top + \frac{1}{\lambda}\boldsymbol{I}\Big)^{-1} \phi(\boldsymbol{x}_t)\boldsymbol{y}_t^\top \boldsymbol{y}_t\phi(\boldsymbol{x}_t)^\top \Big(\Big(\sum_{i=1}^t \phi(\boldsymbol{x}_i)\phi(\boldsymbol{x}_i)^\top + \frac{1}{\lambda}\boldsymbol{I}\Big)^\top\Big)^{-1}\right)
$$

$$
= \|\boldsymbol{y}_t\|_2^2\, \mathrm{Tr}\left(\phi(\boldsymbol{x}_t)\phi(\boldsymbol{x}_t)^\top \Big(\sum_{i=1}^t \phi(\boldsymbol{x}_i)\phi(\boldsymbol{x}_i)^\top + \frac{1}{\lambda}\boldsymbol{I}\Big)^{-1} \phi(\boldsymbol{x}_t)\phi(\boldsymbol{x}_t)^\top \Big(\sum_{i=1}^t \phi(\boldsymbol{x}_i)\phi(\boldsymbol{x}_i)^\top + \frac{1}{\lambda}\boldsymbol{I}\Big)^{-1}\right)
$$

$$
\leq c_y^2\, \mathrm{Tr}\left(\phi(\boldsymbol{x}_t)\phi(\boldsymbol{x}_t)^\top \Big(\sum_{i=1}^t \phi(\boldsymbol{x}_i)\phi(\boldsymbol{x}_i)^\top + \frac{1}{\lambda}\boldsymbol{I}\Big)^{-1} \phi(\boldsymbol{x}_t)\phi(\boldsymbol{x}_t)^\top \Big(\sum_{i=1}^t \phi(\boldsymbol{x}_i)\phi(\boldsymbol{x}_i)^\top + \frac{1}{\lambda}\boldsymbol{I}\Big)^{-1}\right)
$$

$$
= c_y^2\, \mathrm{Tr}\left(\underbrace{\Big(\sum_{i=1}^t \phi(\boldsymbol{x}_i)\phi(\boldsymbol{x}_i)^\top + \frac{1}{\lambda}\boldsymbol{I}\Big)^{-1} \phi(\boldsymbol{x}_t)\phi(\boldsymbol{x}_t)^\top}_{\preceq \frac{1}{t}\boldsymbol{I}} \underbrace{\Big(\sum_{i=1}^t \phi(\boldsymbol{x}_i)\phi(\boldsymbol{x}_i)^\top + \frac{1}{\lambda}\boldsymbol{I}\Big)^{-1} \phi(\boldsymbol{x}_t)\phi(\boldsymbol{x}_t)^\top}_{\preceq \frac{1}{t}\boldsymbol{I}}\right)
$$

$$
\leq c_y^2 D\frac{1}{t^2} \qquad \text{(Assumption 1)}
$$

$$
\leq c_y^2 D\lambda\frac{1}{t}. \qquad (\lambda \geq 1 \geq \tfrac{1}{t} \text{ for all } t = 1, 2, \dots)
$$

For the second term, we have

$$
\begin{aligned}
\frac{1}{t} \operatorname{Tr}\left(\phi(\boldsymbol{x}_t)\boldsymbol{y}_t^\top \boldsymbol{W}^{(t)\top}\right) &= \operatorname{Tr}\left(\phi(\boldsymbol{x}_t)\boldsymbol{y}_t^\top \left(\sum_{i=1}^{t-1} \boldsymbol{y}_i \phi(\boldsymbol{x}_i)^\top\right)\left(\sum_{i=1}^{t-1} \phi(\boldsymbol{x}_i)\phi(\boldsymbol{x}_i)^\top + \frac{1}{\lambda}\boldsymbol{I}\right)^{-1}\right) \\
&= \frac{1}{t} \operatorname{Tr}\left(\left(\sum_{i=1}^{t-1} \phi(\boldsymbol{x}_t)\boldsymbol{y}_t^\top \boldsymbol{y}_i \phi(\boldsymbol{x}_i)^\top\right)\left(\sum_{i=1}^{t-1} \phi(\boldsymbol{x}_i)\phi(\boldsymbol{x}_i)^\top + \frac{1}{\lambda}\boldsymbol{I}\right)^{-1}\right) \\
&\leq c_y^2 \frac{1}{t} \operatorname{Tr}\left(\left(\sum_{i=1}^{t-1} \phi(\boldsymbol{x}_t)\phi(\boldsymbol{x}_i)^\top\right)\left(\sum_{i=1}^{t-1} \phi(\boldsymbol{x}_i)\phi(\boldsymbol{x}_i)^\top + \frac{1}{\lambda}\boldsymbol{I}\right)^{-1}\right) \\
&= c_y^2 \frac{1}{t} \operatorname{Tr}\left(\left(\left(\sum_{i=1}^{t-1} \phi(\boldsymbol{x}_i)\phi(\boldsymbol{x}_i)^\top + \frac{1}{\lambda}\boldsymbol{I}\right)^\top\right)^{-1}\left(\sum_{i=1}^{t-1} \phi(\boldsymbol{x}_t)\phi(\boldsymbol{x}_i)^\top\right)^\top\right) \\
&= c_y^2 \frac{1}{t} \operatorname{Tr}\left(\left(\sum_{i=1}^{t-1} \phi(\boldsymbol{x}_i)\phi(\boldsymbol{x}_i)^\top + \frac{1}{\lambda}\boldsymbol{I}\right)^{-1}\left(\sum_{i=1}^{t-1} \phi(\boldsymbol{x}_i)\phi(\boldsymbol{x}_t)^\top\right)\right) \\
&\leq c_y^2 \frac{1}{t} D. \qquad\qquad\text{(Assumption 1)}
\end{aligned}
$$

Combining the above results, we obtain

$$
\begin{aligned}
|f_t(\boldsymbol{W}^{(t+1)}) - f_t(\boldsymbol{W}^{(t)})| &\leq \operatorname{Tr}\left(\phi(\boldsymbol{x}_t)\phi(\boldsymbol{x}_t)^\top \Delta_t \Delta_t^\top\right) + \frac{4}{t}\operatorname{Tr}\left(\phi(\boldsymbol{x}_t)\boldsymbol{y}_t^\top \boldsymbol{W}^{(t)\top}\right) \\
&\leq c_y^2 D\lambda \frac{1}{t} + 4c_y^2 D\frac{1}{t} \\
&\leq c_y^2 D\lambda \frac{1}{t} + 4\lambda c_y^2 D\frac{1}{t} \qquad\qquad (\lambda \geq 1) \\
&= 5\lambda c_y^2 D\frac{1}{t}.
\end{aligned}
$$

$\square$

**Lemma 4.** *Let $\boldsymbol{W}^{(1)}, \boldsymbol{W}^{(2)}, \ldots$ be the sequence produced by Eq. (12) (or equivalently Eq. (3)). For all $t \geq 1$ and all $\xi \in \mathbb{R}^{D\times S}$ we have*

$$
\sum_{t=1}^T f_t(\boldsymbol{W}^{(t+1)}) \leq \sum_{t=1}^T f_t(\xi)
$$

*Proof.* We prove this lemma by induction. For $T = 1$, it follows directly from the definition of $\boldsymbol{W}^{(t+1)}$, i.e., $\boldsymbol{W}^{(2)} = \arg\min_{\boldsymbol{W}\in\mathbb{R}^{D\times S}} f_1(\boldsymbol{W})$, that

$$
f_1(\boldsymbol{W}^{(2)}) \leq f_1(\xi).
$$

Assume that the inequality holds for $T - 1$, then for all $\xi \in \mathbb{R}^{D\times S}$ we have

$$
\sum_{t=1}^{T-1} f_t(\boldsymbol{W}^{(t+1)}) \leq \sum_{t=1}^{T-1} f_t(\xi).
$$

Adding $f_T(\boldsymbol{W}^{(T+1)})$ to both sides we obtain

$$
\sum_{t=1}^T f_t(\boldsymbol{W}^{(t+1)}) \leq f_T(\boldsymbol{W}^{(T+1)}) + \sum_{t=1}^{T-1} f_t(\xi)
$$

The above holds for all $\xi$ and in particular for $\xi = \boldsymbol{W}^{(T+1)}$. Thus,

$$\sum_{t=1}^{T} f_t(\boldsymbol{W}^{(t+1)}) \le \sum_{t=1}^{T} f_t(\boldsymbol{W}^{(T+1)}) = \min_{\boldsymbol{W} \in \mathbb{R}^{D \times S}} \sum_{t=1}^{T} f_t(\boldsymbol{W})$$

$\square$

**Lemma 5.** *Let* $\boldsymbol{W}^{(1)}, \boldsymbol{W}^{(2)}, \ldots$ *be the sequence of vectors produced by Eq.* (12) *(or equivalently Eq.* (3)*). Then, for all* $t \ge 1$ *and all* $\xi \in \mathbb{R}^{D \times S}$ *we have*

$$\sum_{t=1}^{T} (f_t(\boldsymbol{W}^{(t)}) - f_t(\xi)) \le \ell_{reg}(\xi) - \ell_{reg}(\boldsymbol{W}^{(1)}) + \sum_{t=1}^{T} \left( f_t(\boldsymbol{W}^{(t)}) - f_t(\boldsymbol{W}^{(t+1)}) \right)$$

*Proof.* Let $f_0(\boldsymbol{W}) = \ell_{\text{reg}}(\boldsymbol{W})$. Using Lemma 4 we obtain

$$\sum_{t=0}^{T} f_t(\boldsymbol{W}^{(t+1)}) \le \sum_{t=0}^{T} f_t(\xi). \tag{14}$$

Adding $\sum_{t=0}^{T} f_t(\boldsymbol{W}^{(t)})$ to both sides and rearrange, we get

$$\sum_{t=0}^{T} f_t(\boldsymbol{W}^{(t)}) - \sum_{t=0}^{T} f_t(\xi) \le \sum_{t=0}^{T} f_t(\boldsymbol{W}^{(t)}) - \sum_{t=0}^{T} f_t(\boldsymbol{W}^{(t+1)}).$$

$\square$

### E.2.2   REGRET BOUND FOR ONLINE MODEL LEARNING

Now, we are ready to prove the regret bound for the online model learning problem Eq. (3):

**Corollary 1** (Regret Bound for Online Model Learning). *Let* $\boldsymbol{W}^{(1)}, \boldsymbol{W}^{(2)}, \ldots$ *be the sequence produced by Eq.* (3) *(or equivalently Eq.* (12)*). Suppose that there exist* $\lambda \ge 1, c_W, c_y > 0$ *such that Assumptions 1 and 2 hold. Then, for all* $T \ge 1$ *and all* $\xi \in \mathbb{R}^{D \times S}$*, we have*

$$\sum_{t=1}^{T} f_t(\boldsymbol{W}^{(t)}) - \sum_{t=1}^{T} f_t(\xi) \le \frac{1}{\lambda} c_W^2 + 5\lambda c_y^2 D(\log(T) + 1).$$

*Proof.*

$$\sum_{t=1}^{T} f_t(\boldsymbol{W}^{(t)}) - \sum_{t=1}^{T} f_t(\xi) \le \ell_{\text{reg}}(\xi) - \ell_{\text{reg}}(\boldsymbol{W}^{(1)}) + \sum_{t=1}^{T} \left( f_t(\boldsymbol{W}^{(t)}) - f_t(\boldsymbol{W}^{(t+1)}) \right) \qquad \text{(Lemma 5)}$$

$$\le \frac{1}{\lambda} c_W^2 + \sum_{t=1}^{T} 5c_y^2 D\lambda \frac{1}{t} \qquad \text{(Lemma 3)}$$

$$\le \frac{1}{\lambda} c_W^2 + 5\lambda c_y^2 D(\log(T) + 1). \qquad (\sum_{t=1}^{T} \frac{1}{t} \le \log(T) + 1)$$

$\square$

### E.3   Theoretical analysis for sparse online model learning

### E.3.1   USEFUL LEMMAS

**Lemma 6** (Schur Complement for Block Matrix Inversion (Zhang, 2006)). *Given a matrix* $\boldsymbol{M}$ *that can be partitioned into four blocks as below,*

$$\boldsymbol{M} = \begin{pmatrix} \boldsymbol{M}_{11} & \boldsymbol{M}_{12} \\ \boldsymbol{M}_{21} & \boldsymbol{M}_{22} \end{pmatrix}$$

*it can be inverted blockwise as:*

$$M^{-1} = \begin{pmatrix} M_{11}^{-1} + M_{11}^{-1} M_{12} S^{-1} M_{21} M_{11}^{-1} & -M_{11}^{-1} M_{12} S^{-1} \\ -S^{-1} M_{21} M_{11}^{-1} & S^{-1} \end{pmatrix},$$

*where $S = M_{22} - M_{21} M_{11}^{-1} M_{12}$ is the Schur complement.*

**Proposition 1.** *Given $A, B, W^{(t)}$ as defined by $A = \Phi_{t-1}^\top \Phi_{t-1}$, $B = \Phi_{t-1}^\top Y_{t-1}$, and $W^{(t)} = \left(A + \frac{1}{\lambda} I\right)^{-1} B$ and they can be decomposed into*

$$A = \begin{pmatrix} A_{\bar{s}\bar{s}} & A_{\bar{s}s} \\ A_{s\bar{s}} & A_{ss} \end{pmatrix}, B = \begin{pmatrix} B_{\bar{s}} \\ B_s \end{pmatrix}, W^{(t)} = \begin{pmatrix} W_{\bar{s}}^{(t)} \\ W_s^{(t)} \end{pmatrix},$$

*then, we have*

$$W_{\bar{s}}^{(t)} = \left( \left(A_{\bar{s}\bar{s}} + \frac{1}{\lambda} I\right)^{-1} + D \right) B_{\bar{s}} - \left(A_{\bar{s}\bar{s}} + \frac{1}{\lambda} I\right)^{-1} A_{\bar{s}s} S^{-1} B_s,$$

$$W_s^{(t)} = S^{-1} B_s - S^{-1} A_{s\bar{s}} \left(A_{\bar{s}\bar{s}} + \frac{1}{\lambda} I\right)^{-1} B_{\bar{s}},$$

*where $S := A_{ss} + \frac{1}{\lambda} I - A_{s\bar{s}} \left(A_{\bar{s}\bar{s}} + \frac{1}{\lambda} I\right)^{-1} A_{\bar{s}s}$, $D := (A_{\bar{s}\bar{s}} + \frac{1}{\lambda} I)^{-1} A_{\bar{s}s} S^{-1} A_{s\bar{s}} (A_{\bar{s}\bar{s}} + \frac{1}{\lambda} I)^{-1}$.*

*Proof.* Consider the inverse of $A + \frac{1}{\lambda} I$, which can be written as:

$$A + \frac{1}{\lambda} I = \begin{pmatrix} A_{\bar{s}\bar{s}} + \frac{1}{\lambda} I & A_{\bar{s}s} \\ A_{s\bar{s}} & A_{ss} + \frac{1}{\lambda} I \end{pmatrix}$$

Applying Lemma 6 to our matrix $A + \frac{1}{\lambda} I$, we get:

$$\left(A + \frac{1}{\lambda} I\right)^{-1} = \begin{pmatrix} (A_{\bar{s}\bar{s}} + \frac{1}{\lambda} I)^{-1} + (A_{\bar{s}\bar{s}} + \frac{1}{\lambda} I)^{-1} A_{\bar{s}s} S^{-1} A_{s\bar{s}} (A_{\bar{s}\bar{s}} + \frac{1}{\lambda} I)^{-1} & -(A_{\bar{s}\bar{s}} + \frac{1}{\lambda} I)^{-1} A_{\bar{s}s} S^{-1} \\ -S^{-1} A_{s\bar{s}} (A_{\bar{s}\bar{s}} + \frac{1}{\lambda} I)^{-1} & S^{-1} \end{pmatrix},$$

where $S = A_{ss} + \frac{1}{\lambda} I - A_{s\bar{s}} (A_{\bar{s}\bar{s}} + \frac{1}{\lambda} I)^{-1} A_{\bar{s}s}$.

Now, we can write the expression for $W^{(t)}$ as:

$$W^{(t)} = \begin{pmatrix} (A_{\bar{s}\bar{s}} + \frac{1}{\lambda} I)^{-1} + (A_{\bar{s}\bar{s}} + \frac{1}{\lambda} I)^{-1} A_{\bar{s}s} S^{-1} A_{s\bar{s}} (A_{\bar{s}\bar{s}} + \frac{1}{\lambda} I)^{-1} & -(A_{\bar{s}\bar{s}} + \frac{1}{\lambda} I)^{-1} A_{\bar{s}s} S^{-1} \\ -S^{-1} A_{s\bar{s}} (A_{\bar{s}\bar{s}} + \frac{1}{\lambda} I)^{-1} & S^{-1} \end{pmatrix} \begin{pmatrix} B_{\bar{s}} \\ B_s \end{pmatrix}$$

Therefore, we get

$$W_{\bar{s}}^{(t)} = \left( (A_{\bar{s}\bar{s}} + \frac{1}{\lambda} I)^{-1} + (A_{\bar{s}\bar{s}} + \frac{1}{\lambda} I)^{-1} A_{\bar{s}s} S^{-1} A_{s\bar{s}} (A_{\bar{s}\bar{s}} + \frac{1}{\lambda} I)^{-1} \right) B_{\bar{s}} - (A_{\bar{s}\bar{s}} + \frac{1}{\lambda} I)^{-1} A_{\bar{s}s} S^{-1} B_s,$$

$$W_s^{(t)} = -S^{-1} A_{s\bar{s}} (A_{\bar{s}\bar{s}} + \frac{1}{\lambda} I)^{-1} B_{\bar{s}} + S^{-1} B_s.$$

Let $D := (A_{\bar{s}\bar{s}} + \frac{1}{\lambda} I)^{-1} A_{\bar{s}s} S^{-1} A_{s\bar{s}} (A_{\bar{s}\bar{s}} + \frac{1}{\lambda} I)^{-1}$, we obtain:

$$W_{\bar{s}}^{(t)} = \left( \left(A_{\bar{s}\bar{s}} + \frac{1}{\lambda} I\right)^{-1} + D \right) B_{\bar{s}} - (A_{\bar{s}\bar{s}} + \frac{1}{\lambda} I)^{-1} A_{\bar{s}s} S^{-1} B_s,$$

$$W_s^{(t)} = S^{-1} B_s - S^{-1} A_{s\bar{s}} \left(A_{\bar{s}\bar{s}} + \frac{1}{\lambda} I\right)^{-1} B_{\bar{s}}.$$

$\square$

**Proposition 2.** *Suppose that Assumption 3 holds, then we have*

$$\left( \begin{matrix} \mathbf{0} \\ \left( A_{ss}^{(t)} + \frac{1}{\lambda}I \right)^{-1} B_s^{(t)} \end{matrix} \right) \preceq K \left( A^{(t)} + \frac{1}{\lambda}I \right)^{-1} \phi(x_t) y_t^\top.$$

*Proof.* Note that

$$A = \left( \begin{matrix} A_{\bar{s}\bar{s}} & A_{\bar{s}s} \\ A_{s\bar{s}} & A_{ss} \end{matrix} \right), B = \left( \begin{matrix} B_{\bar{s}} \\ B_s \end{matrix} \right).$$

We have

$$\left( A + \frac{1}{\lambda}I \right) \left( \begin{matrix} \mathbf{0} & \mathbf{0} \\ \mathbf{0} & (A_{ss} + \frac{1}{\lambda}I)^{-1} \end{matrix} \right) = \left( \begin{matrix} A_{\bar{s}\bar{s}} + \frac{1}{\lambda}I & A_{\bar{s}s} \\ A_{s\bar{s}} & A_{ss} + \frac{1}{\lambda}I \end{matrix} \right) \left( \begin{matrix} \mathbf{0} & \mathbf{0} \\ \mathbf{0} & (A_{ss} + \frac{1}{\lambda}I)^{-1} \end{matrix} \right)$$

$$= \left( \begin{matrix} \mathbf{0} & A_{\bar{s}s} \left( A_{ss} + \frac{1}{\lambda}I \right)^{-1} \\ \mathbf{0} & I \end{matrix} \right).$$

Then, we have

$$\left( \begin{matrix} \mathbf{0} \\ (A_{ss} + \frac{1}{\lambda}I)^{-1} B_s \end{matrix} \right) = \left( \begin{matrix} \mathbf{0} & \mathbf{0} \\ \mathbf{0} & (A_{ss} + \frac{1}{\lambda}I)^{-1} \end{matrix} \right) \left( \begin{matrix} B_{\bar{s}} \\ B_s \end{matrix} \right)$$

$$= \left( A + \frac{1}{\lambda}I \right)^{-1} \left( \begin{matrix} \mathbf{0} & A_{\bar{s}s} \left( A_{ss} + \frac{1}{\lambda}I \right)^{-1} \\ \mathbf{0} & I \end{matrix} \right) B. \tag{15}$$

Note that by the properties of the block matrix's spectral norm, we get

$$\left\| \left( \begin{matrix} \mathbf{0} & A_{\bar{s}s} \left( A_{ss} + \frac{1}{\lambda}I \right)^{-1} \\ \mathbf{0} & I \end{matrix} \right) \right\|_2 \leq \sqrt{ \left\| A_{\bar{s}s} \left( A_{ss} + \frac{1}{\lambda}I \right)^{-1} \right\|_2^2 + \|I\|_2^2 }$$

$$= \sqrt{ \left\| A_{\bar{s}s} \left( A_{ss} + \frac{1}{\lambda}I \right)^{-1} \right\|_2^2 + 1 } \tag{16}$$

Additionally, if Assumption 1 holds for $c_y \geq 1$, we have for all $x$

$$By_t \phi(x_t)^\top \phi(x_t) y_t^\top = \left( \sum_{i=1}^t \phi(x_i) y_i^\top y_t \phi(x_t)^\top \right) \phi(x_t) y_t^\top$$

$$\preceq c_y^2 \left( \sum_{i=1}^t \phi(x_i) \phi(x_t)^\top \right) \phi(x_t) y_t^\top$$

$$\preceq c_y^2 \left( \sum_{i=1}^t \phi(x_i) \phi(x_i)^\top + \frac{1}{\lambda}I \right) \phi(x_t) y_t^\top \qquad \text{(by Assumption 1)}$$

Moreover, by

$$\left( y_t \phi(x_t)^\top \phi(x_t) y_t^\top \right)^{-1} = \left( \phi(x_t)^\top \phi(x_t) y_t^\top y_t \right)^{-1} y_t y_t^\top$$

we obtain

$$
\begin{aligned}
\boldsymbol{B} &\preceq c_y^2 \left( \sum_{i=1}^{t} \phi(x_i)\phi(x_i)^\top + \frac{1}{\lambda}\boldsymbol{I} \right) \phi(x_t)\boldsymbol{y}_t^\top \left( \boldsymbol{y}_t \phi(x_t)^\top \phi(x_t) \boldsymbol{y}_t^\top \right)^{-1} \\
&= c_y^2 \left( \sum_{i=1}^{t} \phi(x_i)\phi(x_i)^\top + \frac{1}{\lambda}\boldsymbol{I} \right) \phi(x_t)\boldsymbol{y}_t^\top \boldsymbol{y}_t \boldsymbol{y}_t^\top \left( \phi(x_t)^\top \phi(x_t) \boldsymbol{y}_t^\top \boldsymbol{y}_t^\top \right)^{-1} \\
&= \frac{c_y^2}{\phi(x_t)^\top \phi(x_t)} \left( \sum_{i=1}^{t} \phi(x_i)\phi(x_i)^\top + \frac{1}{\lambda}\boldsymbol{I} \right) \phi(x_t)\boldsymbol{y}_t^\top \\
&\preceq K'\phi(x_t)\boldsymbol{y}_t^\top,
\end{aligned}
\tag{17}
$$

where $K' \geq c_y^2 \left( \frac{\sum_{i=1}^{t} \phi(x_i)^\top \phi(x_i)}{\phi(x_t)^\top \phi(x_t)} + 1 \right)$.

Combining Eq. (15) to (17) we obtain,

$$
\begin{aligned}
\begin{pmatrix} \boldsymbol{0} \\ \left(\boldsymbol{A}_{ss} + \frac{1}{\lambda}\boldsymbol{I}\right)^{-1}\boldsymbol{B}_s \end{pmatrix} &= \left(\boldsymbol{A} + \frac{1}{\lambda}\boldsymbol{I}\right)^{-1} \begin{pmatrix} \boldsymbol{0} & \boldsymbol{A}_{\overline{s}s}\left(\boldsymbol{A}_{ss} + \frac{1}{\lambda}\boldsymbol{I}\right)^{-1} \\ \boldsymbol{0} & \boldsymbol{I} \end{pmatrix} \boldsymbol{B} \\
&\preceq \sqrt{ \left\| \boldsymbol{A}_{\overline{s}s}\left(\boldsymbol{A}_{ss} + \frac{1}{\lambda}\boldsymbol{I}\right)^{-1} \right\|_2^2 + 1 } \left(\boldsymbol{A} + \frac{1}{\lambda}\boldsymbol{I}\right)^{-1}\boldsymbol{B} \\
&\preceq K' \sqrt{ \left\| \boldsymbol{A}_{\overline{s}s}\left(\boldsymbol{A}_{ss} + \frac{1}{\lambda}\boldsymbol{I}\right)^{-1} \right\|_2^2 + 1 } \left(\boldsymbol{A} + \frac{1}{\lambda}\boldsymbol{I}\right)^{-1} \phi(x_t)\boldsymbol{y}_t^\top \\
&= K \left(\boldsymbol{A} + \frac{1}{\lambda}\boldsymbol{I}\right)^{-1} \phi(x_t)\boldsymbol{y}_t^\top,
\end{aligned}
$$

where we choose

$$
\begin{aligned}
K &= K' \sqrt{ \left\| \boldsymbol{A}_{\overline{s}s}\left(\boldsymbol{A}_{ss} + \frac{1}{\lambda}\boldsymbol{I}\right)^{-1} \right\|_2^2 + 1 } \\
&\geq c_y^2 \left( \frac{\sum_{i=1}^{t} \phi(x_i)^\top \phi(x_i)}{\phi(x_t)^\top \phi(x_t)} + 1 \right) \sqrt{ \left\| \boldsymbol{A}_{\overline{s}s}\left(\boldsymbol{A}_{ss} + \frac{1}{\lambda}\boldsymbol{I}\right)^{-1} \right\|_2^2 + 1 }.
\end{aligned}
$$

$\square$

**Proposition 3.** *Let $\boldsymbol{W}^{(1)}, \boldsymbol{W}^{(2)}, \ldots$ be the sequence produced by the online model learning Eq. (3) and $\widetilde{\boldsymbol{W}}^{(1)}, \widetilde{\boldsymbol{W}}^{(2)}, \ldots$ be the sequence produced by the sparse model learning Eq. (4). For all $t \geq 1$, let $\boldsymbol{M}^{(t)} := \boldsymbol{A}_{s\overline{s}}^{(t)} \left( \boldsymbol{A}_{\overline{s}\overline{s}}^{(t)} + \frac{1}{\lambda}\boldsymbol{I} \right)^{-1} \boldsymbol{A}_{\overline{s}s}^{(t)}$ and*

$$
r_M := \sup_{t,s \in \{s_1,\ldots,s_t\}} \left\| \left( \boldsymbol{A}_{ss}^{(t)} + \frac{1}{\lambda}\boldsymbol{I} + \boldsymbol{M}^{(t)} \right) \left( \boldsymbol{A}_{ss}^{(t)} + \frac{1}{\lambda}\boldsymbol{I} - \boldsymbol{M}^{(t)} \right)^{-1} \right\|_F
\tag{18}
$$

*Suppose that there exist $\lambda$ such that Assumption 3 holds. Then, for all $t \geq 1$, we have*

$$
\boldsymbol{W}^{(t+1)} - \widetilde{\boldsymbol{W}}^{(t+1)} \preceq K r_M \Delta_t,
$$

*where $\Delta_t$ is defined by Eq. (13) in Lemma 2.*

*Proof.* Note that $\boldsymbol{A}$ is symmetric and $\boldsymbol{A}_{s\overline{s}} = \boldsymbol{A}_{\overline{s}s}^\top$, then, $\boldsymbol{M} := \boldsymbol{A}_{s\overline{s}}\left(\boldsymbol{A}_{\overline{s}\overline{s}} + \frac{1}{\lambda}\boldsymbol{I}\right)^{-1}\boldsymbol{A}_{\overline{s}s} \succeq \boldsymbol{0}$, and we have $\boldsymbol{S} := \boldsymbol{A}_{ss} + \frac{1}{\lambda}\boldsymbol{I} - \boldsymbol{A}_{s\overline{s}}\left(\boldsymbol{A}_{\overline{s}\overline{s}} + \frac{1}{\lambda}\boldsymbol{I}\right)^{-1}\boldsymbol{A}_{\overline{s}s} = \boldsymbol{A}_{ss} + \frac{1}{\lambda}\boldsymbol{I} - \boldsymbol{M}$. Moreover,

$$
\begin{aligned}
\boldsymbol{A}_{s\overline{s}}\boldsymbol{D} &= \boldsymbol{A}_{s\overline{s}}(\boldsymbol{A}_{\overline{s}\overline{s}} + \frac{1}{\lambda}\boldsymbol{I})^{-1}\boldsymbol{A}_{\overline{s}s}\boldsymbol{S}^{-1}\boldsymbol{A}_{s\overline{s}}(\boldsymbol{A}_{\overline{s}\overline{s}} + \frac{1}{\lambda}\boldsymbol{I})^{-1} \\
&= \boldsymbol{M}\boldsymbol{S}^{-1}(\boldsymbol{A}_{\overline{s}\overline{s}} + \frac{1}{\lambda}\boldsymbol{I})^{-1} \succeq \boldsymbol{0}.
\end{aligned}
$$

Then, with Proposition 1, we get

$$
\boldsymbol{W}_s^{(t)} - \left(\boldsymbol{A}_{ss} + \frac{1}{\lambda}\boldsymbol{I}\right)^{-1}\boldsymbol{B}_s + \left(\boldsymbol{A}_{ss} + \frac{1}{\lambda}\boldsymbol{I}\right)^{-1}\boldsymbol{A}_{s\bar{s}}\boldsymbol{W}_{\bar{s}}^{(t)}
$$

$$
= \boldsymbol{S}^{-1}\boldsymbol{B}_s - \boldsymbol{S}^{-1}\boldsymbol{A}_{s\bar{s}}\left(\boldsymbol{A}_{\bar{s}\bar{s}} + \frac{1}{\lambda}\boldsymbol{I}\right)^{-1}\boldsymbol{B}_{\bar{s}} - \left(\boldsymbol{A}_{ss} + \frac{1}{\lambda}\boldsymbol{I}\right)^{-1}\boldsymbol{B}_s
$$

$$
- \left(\boldsymbol{A}_{ss} + \frac{1}{\lambda}\boldsymbol{I}\right)^{-1}\boldsymbol{A}_{s\bar{s}}\left(\left(\left(\boldsymbol{A}_{\bar{s}\bar{s}} + \frac{1}{\lambda}\boldsymbol{I}\right)^{-1} + \boldsymbol{D}\right)\boldsymbol{B}_{\bar{s}} - (\boldsymbol{A}_{\bar{s}\bar{s}} + \frac{1}{\lambda}\boldsymbol{I})^{-1}\boldsymbol{A}_{\bar{s}s}\boldsymbol{S}^{-1}\boldsymbol{B}_s\right)
$$

$$
= \left(\boldsymbol{S}^{-1} + \left(\boldsymbol{A}_{ss} + \frac{1}{\lambda}\boldsymbol{I}\right)^{-1}\boldsymbol{A}_{s\bar{s}}\left(\boldsymbol{A}_{\bar{s}\bar{s}} + \frac{1}{\lambda}\boldsymbol{I}\right)^{-1}\boldsymbol{A}_{\bar{s}s}\boldsymbol{S}^{-1} - \left(\boldsymbol{A}_{ss} + \frac{1}{\lambda}\boldsymbol{I}\right)^{-1}\right)\boldsymbol{B}_s
$$

$$
- \left(\boldsymbol{S}^{-1}\boldsymbol{A}_{s\bar{s}}\left(\boldsymbol{A}_{\bar{s}\bar{s}} + \frac{1}{\lambda}\boldsymbol{I}\right)^{-1} + \left(\boldsymbol{A}_{ss} + \frac{1}{\lambda}\boldsymbol{I}\right)^{-1}\boldsymbol{A}_{s\bar{s}}\left(\left(\boldsymbol{A}_{\bar{s}\bar{s}} + \frac{1}{\lambda}\boldsymbol{I}\right)^{-1} + \boldsymbol{D}\right)\right)\boldsymbol{B}_{\bar{s}}
$$

$$
= \left(\boldsymbol{S}^{-1} + \left(\boldsymbol{A}_{ss} + \frac{1}{\lambda}\boldsymbol{I}\right)^{-1}\boldsymbol{A}_{s\bar{s}}\left(\boldsymbol{A}_{\bar{s}\bar{s}} + \frac{1}{\lambda}\boldsymbol{I}\right)^{-1}\boldsymbol{A}_{\bar{s}s}\boldsymbol{S}^{-1} - \left(\boldsymbol{A}_{ss} + \frac{1}{\lambda}\boldsymbol{I}\right)^{-1}\right)\boldsymbol{B}_s
$$

$$
- \left(\underbrace{\left(\boldsymbol{S}^{-1} + \left(\boldsymbol{A}_{ss} + \frac{1}{\lambda}\boldsymbol{I}\right)^{-1}\right)\boldsymbol{A}_{s\bar{s}}\left(\boldsymbol{A}_{\bar{s}\bar{s}} + \frac{1}{\lambda}\boldsymbol{I}\right)^{-1}}_{\succeq 0} + \underbrace{\left(\boldsymbol{A}_{ss} + \frac{1}{\lambda}\boldsymbol{I}\right)^{-1}\boldsymbol{A}_{s\bar{s}}\boldsymbol{D}}_{\succeq 0}\right)\boldsymbol{B}_{\bar{s}}
$$

$$
\preceq \left(\boldsymbol{S}^{-1} + \left(\boldsymbol{A}_{ss} + \frac{1}{\lambda}\boldsymbol{I}\right)^{-1}\boldsymbol{A}_{s\bar{s}}\left(\boldsymbol{A}_{\bar{s}\bar{s}} + \frac{1}{\lambda}\boldsymbol{I}\right)^{-1}\boldsymbol{A}_{\bar{s}s}\boldsymbol{S}^{-1} - \left(\boldsymbol{A}_{ss} + \frac{1}{\lambda}\boldsymbol{I}\right)^{-1}\right)\boldsymbol{B}_s
$$

$$
= \left(\boldsymbol{S}^{-1} + \left(\boldsymbol{A}_{ss} + \frac{1}{\lambda}\boldsymbol{I}\right)^{-1}\boldsymbol{M}\boldsymbol{S}^{-1} - \left(\boldsymbol{A}_{ss} + \frac{1}{\lambda}\boldsymbol{I}\right)^{-1}\right)\boldsymbol{B}_s.
$$

Let $r_M \coloneqq \sup_{t,s\in\{s_1,\dots,s_t\}}\left\|\left(\boldsymbol{A}_{ss}^{(t)} + \frac{1}{\lambda}\boldsymbol{I} + \boldsymbol{M}^{(t)}\right)\left(\boldsymbol{A}_{ss}^{(t)} + \frac{1}{\lambda}\boldsymbol{I} - \boldsymbol{M}^{(t)}\right)^{-1}\right\|_F$, we have

$$
\boldsymbol{W}_s^{(t)} - \left(\boldsymbol{A}_{ss} + \frac{1}{\lambda}\boldsymbol{I}\right)^{-1}\boldsymbol{B}_s + \left(\boldsymbol{A}_{ss} + \frac{1}{\lambda}\boldsymbol{I}\right)^{-1}\boldsymbol{A}_{s\bar{s}}\boldsymbol{W}_{\bar{s}}^{(t)}
$$

$$
\preceq \left(\boldsymbol{S}^{-1} + \left(\boldsymbol{A}_{ss} + \frac{1}{\lambda}\boldsymbol{I}\right)^{-1}\boldsymbol{M}\boldsymbol{S}^{-1} - \left(\boldsymbol{A}_{ss} + \frac{1}{\lambda}\boldsymbol{I}\right)^{-1}\right)\boldsymbol{B}_s
$$

$$
= \left(\left(\boldsymbol{I} + \left(\boldsymbol{A}_{ss} + \frac{1}{\lambda}\boldsymbol{I}\right)^{-1}\boldsymbol{M}\right)\left(\boldsymbol{A}_{ss} + \frac{1}{\lambda}\boldsymbol{I} - \boldsymbol{M}\right)^{-1} - \left(\boldsymbol{A}_{ss} + \frac{1}{\lambda}\boldsymbol{I}\right)^{-1}\right)\boldsymbol{B}_s
$$

$$
= \left(\left(\boldsymbol{A}_{ss} + \frac{1}{\lambda}\boldsymbol{I}\right)^{-1}\underbrace{\left(\boldsymbol{A}_{ss} + \frac{1}{\lambda}\boldsymbol{I} + \boldsymbol{M}\right)\left(\boldsymbol{A}_{ss} + \frac{1}{\lambda}\boldsymbol{I} - \boldsymbol{M}\right)^{-1}}_{\preceq r_M \boldsymbol{I}} - \left(\boldsymbol{A}_{ss} + \frac{1}{\lambda}\boldsymbol{I}\right)^{-1}\right)\boldsymbol{B}_s
$$

$$
= \left(\boldsymbol{A}_{ss} + \frac{1}{\lambda}\boldsymbol{I}\right)^{-1}\underbrace{\left(\left(\boldsymbol{A}_{ss} + \frac{1}{\lambda}\boldsymbol{I} + \boldsymbol{M}\right)\left(\boldsymbol{A}_{ss} + \frac{1}{\lambda}\boldsymbol{I} - \boldsymbol{M}\right)^{-1} - \boldsymbol{I}\right)}_{\preceq (r_M - 1)\boldsymbol{I}}\boldsymbol{B}_s
$$

$$
\preceq (r_M - 1)\cdot\left(\boldsymbol{A}_{ss} + \frac{1}{\lambda}\boldsymbol{I}\right)^{-1}\boldsymbol{B}_s
$$

Recall from Assumption 3 and Proposition 2 that

$$\begin{pmatrix} \mathbf{0} \\ \left(\mathbf{A}_{ss}^{(t)} + \frac{1}{\lambda}\mathbf{I}\right)^{-1}\mathbf{B}_s^{(t)} \end{pmatrix} \preceq K\left(\mathbf{A}^{(t)} + \frac{1}{\lambda}\mathbf{I}\right)^{-1}\phi(x_t)\mathbf{y}_t^\top,$$

which gives that

$$\begin{aligned}
&\mathbf{W}^{(t+1)} - \widetilde{\mathbf{W}}^{(t+1)} \\
&= \mathbf{W}^{(t+1)} - \left[\mathbf{W}_{\overline{s}}^{(t+1)}; \mathbf{W}_s^{(t+1)}\right] \\
&= \mathbf{W}^{(t+1)} - \left[\mathbf{W}_{\overline{s}}^{(t)}; \mathbf{W}_s^{(t+1)}\right] \\
&\preceq \mathbf{W}^{(t)} + \Delta_t - \left[\mathbf{W}_{\overline{s}}^{(t)}; \mathbf{W}_s^{(t+1)}\right] \\
&= \mathbf{W}^{(t)} + \Delta_t - \left[\mathbf{W}_{\overline{s}}^{(t)}; \left(\mathbf{A}_{ss}^{(t)} + \frac{1}{\lambda}\mathbf{I}\right)^{-1}\left(\mathbf{B}_s^{(t)} - \mathbf{A}_{s\overline{s}}^{(t)}\mathbf{W}_{\overline{s}}^{(t)}\right)\right] \\
&= \left[\mathbf{0}; \mathbf{W}_s^{(t)} - \left(\mathbf{A}_{ss}^{(t)} + \frac{1}{\lambda}\mathbf{I}\right)^{-1}\mathbf{B}_s^{(t)} + \left(\mathbf{A}_{ss}^{(t)} + \frac{1}{\lambda}\mathbf{I}\right)^{-1}\mathbf{A}_{s\overline{s}}^{(t)}\mathbf{W}_{\overline{s}}^{(t)}\right] + \Delta_t \\
&\preceq \left[\mathbf{0}; (r_M - 1)\left(\mathbf{A}_{ss}^{(t)} + \frac{1}{\lambda}\mathbf{I}\right)^{-1}\mathbf{B}_s^{(t)}\right] + \Delta_t \\
&\preceq K(r_M - 1)\Delta_t + \Delta_t \qquad\qquad\qquad\qquad\qquad\qquad \text{(Assumption 3 and Proposition 2)} \\
&\preceq Kr_M\Delta_t.
\end{aligned}$$

$\square$

### E.3.2 REGRET BOUNDS FOR SPARSE ONLINE MODEL LEARNING

**Theorem 1** (Regret Bounds for Sparse Online Model Learning). *Let $\widetilde{\mathbf{W}}^{(1)}, \widetilde{\mathbf{W}}^{(2)}, \ldots$ be the sequence produced by the sparse online model learning Eq. (4). Let $r_M \geq 1$ be a constant defined as in Eq. (18). Suppose that there exist $c_W, c_y > 0$ and $\lambda$ such that Assumptions 1 to 3 hold. Then, for all $T \geq 1$ and all $\xi \in \mathbb{R}^{D\times S}$, we have*

$$\text{Regret}(T) := \sum_{t=1}^T f_t(\widetilde{\mathbf{W}}^{(t)}) - \sum_{t=1}^T f_t(\xi) \leq \frac{1}{\lambda}c_W^2 + 5\lambda(K^2 r_M^2 + 1)c_y^2 D(\log(T) + 1),$$

*Furthermore, If $\lambda = c_W/c_y\sqrt{5(K^2 r_M^2 + 1)D(\log(T) + 1)} \geq 1$ and it satisfies Assumptions 1 and 3, then we have*

$$\text{Regret}(T) \leq c_W c_y\sqrt{20(K^2 r_M^2 + 1)D(\log(T) + 1)}.$$

*Proof.* For $t = 1$, we have $f_t(\widetilde{\mathbf{W}}^{(1)}) = f_t(\mathbf{W}^{(1)})$ and $f_t(\widetilde{\mathbf{W}}^{(1)}) - f_t(\mathbf{W}^{(1)}) = 0$.

For $t \geq 2$, denote by $r_M' := Kr_M$, we have

$$
\begin{aligned}
& f_t(\widetilde{\boldsymbol{W}}^{(t)}) - f_t(\boldsymbol{W}^{(t)}) \\
& = \operatorname{Tr}(\widetilde{\boldsymbol{W}}^{(t)\top}\phi(\boldsymbol{x}_t)\phi(\boldsymbol{x}_t)^\top\widetilde{\boldsymbol{W}}^{(t)}) + \operatorname{Tr}(\boldsymbol{y}_t^\top\boldsymbol{y}_t) - 2\operatorname{Tr}(\widetilde{\boldsymbol{W}}^{(t)\top}\phi(\boldsymbol{x}_t)\boldsymbol{y}_t^\top) \\
& \quad - \operatorname{Tr}(\boldsymbol{W}^{(t)\top}\phi(\boldsymbol{x}_t)\phi(\boldsymbol{x}_t)^\top\boldsymbol{W}^{(t)}) - \operatorname{Tr}(\boldsymbol{y}_t^\top\boldsymbol{y}_t) + 2\operatorname{Tr}(\boldsymbol{W}^{(t)\top}\phi(\boldsymbol{x}_t)\boldsymbol{y}_t^\top) \\
& = \operatorname{Tr}\left(\phi(\boldsymbol{x}_t)\phi(\boldsymbol{x}_t)^\top\widetilde{\boldsymbol{W}}^{(t)}\widetilde{\boldsymbol{W}}^{(t)\top}\right) - 2\operatorname{Tr}\left(\phi(\boldsymbol{x}_t)\boldsymbol{y}_t^\top\widetilde{\boldsymbol{W}}^{(t)\top}\right) \\
& \quad - \operatorname{Tr}\left(\phi(\boldsymbol{x}_t)\phi(\boldsymbol{x}_t)^\top\boldsymbol{W}^{(t)}\boldsymbol{W}^{(t)\top}\right) + 2\operatorname{Tr}\left(\phi(\boldsymbol{x}_t)\boldsymbol{y}_t^\top\boldsymbol{W}^{(t)\top}\right) \\
& \leq \left|\operatorname{Tr}\left(\phi(\boldsymbol{x}_t)\phi(\boldsymbol{x}_t)^\top\left(\boldsymbol{W}^{(t)} - \widetilde{\boldsymbol{W}}^{(t)}\right)\left(\boldsymbol{W}^{(t)} - \widetilde{\boldsymbol{W}}^{(t)}\right)^\top\right)\right| + 2\left|\operatorname{Tr}\left(\phi(\boldsymbol{x}_t)\boldsymbol{y}_t^\top\left(\boldsymbol{W}^{(t)} - \widetilde{\boldsymbol{W}}^{(t)}\right)^\top\right)\right| \\
& \leq r_M'^2 \cdot \operatorname{Tr}\left(\phi(\boldsymbol{x}_t)\phi(\boldsymbol{x}_t)^\top\Delta_{t-1}\Delta_{t-1}^\top\right) + 2r_M' \cdot \operatorname{Tr}\left(\phi(\boldsymbol{x}_t)\boldsymbol{y}_t^\top\Delta_{t-1}^\top\right) \qquad \text{(Proposition 3)}
\end{aligned}
$$

For the first term in the right hand side of the above inequality, we have

$$
\begin{aligned}
& \operatorname{Tr}\left(\phi(\boldsymbol{x}_t)\phi(\boldsymbol{x}_t)^\top\Delta_{t-1}\Delta_{t-1}^\top\right) \\
& = \operatorname{Tr}\left(\phi(\boldsymbol{x}_t)\phi(\boldsymbol{x}_t)^\top\left(\sum_{i=1}^{t-1}\phi(\boldsymbol{x}_i)\phi(\boldsymbol{x}_i)^\top + \frac{1}{\lambda}\boldsymbol{I}\right)^{-1}\phi(\boldsymbol{x}_{t-1})\boldsymbol{y}_{t-1}^\top\boldsymbol{y}_{t-1}\phi(\boldsymbol{x}_{t-1})^\top\left(\left(\sum_{i=1}^{t-1}\phi(\boldsymbol{x}_i)\phi(\boldsymbol{x}_i)^\top + \frac{1}{\lambda}\boldsymbol{I}\right)^\top\right)^{-1}\right) \\
& = \|\boldsymbol{y}_{t-1}\|_2^2 \operatorname{Tr}\left(\phi(\boldsymbol{x}_t)\phi(\boldsymbol{x}_t)^\top\left(\sum_{i=1}^{t-1}\phi(\boldsymbol{x}_i)\phi(\boldsymbol{x}_i)^\top + \frac{1}{\lambda}\boldsymbol{I}\right)^{-1}\phi(\boldsymbol{x}_{t-1})\phi(\boldsymbol{x}_{t-1})^\top\left(\sum_{i=1}^{t-1}\phi(\boldsymbol{x}_i)\phi(\boldsymbol{x}_i)^\top + \frac{1}{\lambda}\boldsymbol{I}\right)^{-1}\right) \\
& \leq c_y^2 \operatorname{Tr}\left(\phi(\boldsymbol{x}_t)\phi(\boldsymbol{x}_t)^\top\left(\sum_{i=1}^{t-1}\phi(\boldsymbol{x}_i)\phi(\boldsymbol{x}_i)^\top + \frac{1}{\lambda}\boldsymbol{I}\right)^{-1}\phi(\boldsymbol{x}_{t-1})\phi(\boldsymbol{x}_{t-1})^\top\left(\sum_{i=1}^{t-1}\phi(\boldsymbol{x}_i)\phi(\boldsymbol{x}_i)^\top + \frac{1}{\lambda}\boldsymbol{I}\right)^{-1}\right) \\
& = c_y^2 \operatorname{Tr}\left(\underbrace{\left(\sum_{i=1}^{t-1}\phi(\boldsymbol{x}_i)\phi(\boldsymbol{x}_i)^\top + \frac{1}{\lambda}\boldsymbol{I}\right)^{-1}\phi(\boldsymbol{x}_{t-1})\phi(\boldsymbol{x}_{t-1})^\top}_{\preceq\frac{1}{t-1}\boldsymbol{I}}\underbrace{\left(\sum_{i=1}^{t-1}\phi(\boldsymbol{x}_i)\phi(\boldsymbol{x}_i)^\top + \frac{1}{\lambda}\boldsymbol{I}\right)^{-1}\phi(\boldsymbol{x}_t)\phi(\boldsymbol{x}_t)^\top}_{\preceq\frac{1}{t-1}\boldsymbol{I}}\right) \\
& \leq c_y^2 D \frac{1}{(t-1)^2} \qquad\qquad\qquad\qquad\qquad\qquad\qquad\qquad\qquad \text{(Assumption 1)} \\
& \leq c_y^2 D\lambda \frac{1}{t-1}. \qquad\qquad\qquad\qquad\qquad\qquad\qquad (\lambda \geq 1 \geq \tfrac{1}{t-1} \text{ for all } t = 2,\dots)
\end{aligned}
$$

For the last term, we have

$$
\begin{aligned}
\operatorname{Tr}\left(\phi(\boldsymbol{x}_t)\boldsymbol{y}_t^\top\Delta_{t-1}^\top\right) & = \operatorname{Tr}\left(\phi(\boldsymbol{x}_t)\boldsymbol{y}_t^\top\boldsymbol{y}_{t-1}\phi(\boldsymbol{x}_{t-1})^\top\left(\left(\sum_{i=1}^{t-1}\phi(\boldsymbol{x}_i)\phi(\boldsymbol{x}_i)^\top + \frac{1}{\lambda}\boldsymbol{I}\right)^\top\right)^{-1}\right) \\
& \leq c_y^2 \operatorname{Tr}\left(\phi(\boldsymbol{x}_t)\phi(\boldsymbol{x}_{t-1})^\top\left(\sum_{i=1}^{t-1}\phi(\boldsymbol{x}_i)\phi(\boldsymbol{x}_i)^\top + \frac{1}{\lambda}\boldsymbol{I}\right)^{-1}\right) \\
& = c_y^2 \operatorname{Tr}\left(\underbrace{\left(\sum_{i=1}^{t-1}\phi(\boldsymbol{x}_i)\phi(\boldsymbol{x}_i)^\top + \frac{1}{\lambda}\boldsymbol{I}\right)^{-1}\phi(\boldsymbol{x}_t)\phi(\boldsymbol{x}_{t-1})^\top}_{\preceq\frac{1}{t-1}\boldsymbol{I}}\right) \\
& \leq c_y^2 D \frac{1}{t-1}.
\end{aligned}
$$

Therefore, the regret bound is upper-bounded as below:

$$
\begin{aligned}
\text{Regret}(T) &= \sum_{t=1}^{T} f_t(\widetilde{\boldsymbol{W}}^{(t)}) - \sum_{t=1}^{T} f_t(\boldsymbol{W}^{(t)}) + \sum_{t=1}^{T} f_t(\boldsymbol{W}^{(t)}) - \sum_{t=1}^{T} f_t(\xi) \\
&\leq \sum_{t=1}^{T} \left( f_t(\widetilde{\boldsymbol{W}}^{(t)}) - f_t(\boldsymbol{W}^{(t)}) \right) + \frac{1}{\lambda} c_W^2 + 5\lambda c_y^2 D(\log(T) + 1) \qquad\qquad \text{(Corollary 1)} \\
&\leq c_y^2 D \left( r_M'^2 \lambda + 2r_M' \right) \sum_{t=2}^{T} \frac{1}{t-1} + \frac{1}{\lambda} c_W^2 + 5\lambda c_y^2 D(\log(T) + 1) \\
&\leq c_y^2 D \left( r_M'^2 \lambda + 2r_M' \right) (\log(T-1) + 1) + \frac{1}{\lambda} c_W^2 + 5\lambda c_y^2 D(\log(T) + 1) \\
&\leq \frac{1}{\lambda} c_W^2 + 5\lambda (r_M'^2 + 1) c_y^2 D(\log(T) + 1), \qquad\qquad\qquad\qquad (\lambda \geq 1, r_M' \geq 1)
\end{aligned}
$$

which holds for all $\lambda \geq 1$ that satisfies Assumptions 1 and 3.

Furthermore, If $c_W \geq c_y \sqrt{5(K^2 r_M^2 + 1)D(\log(T) + 1)}$, and $\lambda = \sqrt{\frac{c_W^2}{5(K^2 r_M^2 + 1)c_y^2 D(\log(T)+1)}}$ satisfies Assumptions 1 and 3, then we have

$$
\text{Regret}(T) \leq c_W c_y \sqrt{20(K^2 r_M^2 + 1)D(\log(T) + 1)}.
$$

$\qquad\qquad\qquad\qquad\qquad\qquad\qquad\qquad\qquad\qquad\qquad\qquad\qquad\qquad\qquad\qquad\qquad\qquad \square$

