# OpenReview forum: "Continual Reinforcement Learning by Planning with Online World Models"
_ICML.cc/2025/Conference — ICML 2025 spotlightposter_

### Official Review · Reviewer_WQD5 · 2025-02-22

**Overall Recommendation:** 4

**Summary:**

In this work the authors propose a new task-unknown continual reinforcement learning setting, in which the agent needs to learn a sequence of tasks without exactly boundary or id. To deal with this new setting, the authors propose a new continual RL method, OA, which introduce a sparse world model by FTL models to overcome the catastrophic forgetting. Besides, the authors propose a new CRL benchmark, the Continual Bench, which is helpful for following works

**Claims And Evidence:**

I think the claims are clear and convincing evidence.

**Essential References Not Discussed:**

N/A

**Experimental Designs Or Analyses:**

I would like to know that why the horizon $H$ is set to 15? As I know, this is a large number for the model-based RL, but not an enough number when you want to generate a whole trajectories.

**Methods And Evaluation Criteria:**

I have two questions about the algorithm in Section A.2.

First, in the line 2, the loop start with a condition: task changes. How does the agent know when the task changes? What exactly this condition sentence means?

Second, in line 12 to line 13, the $A_{ss}^{\left(t\right)}$ is calculated with $A_{ss}^{\left(t+1\right)}$ and $\phi_s$. Will the sparsity of $A_{ss}$ and $B_{s}$ change after these lines? How to maintain the sparsity?

**Other Comments Or Suggestions:**

N/A

**Other Strengths And Weaknesses:**

N/A

**Questions For Authors:**

I’m sorry but I do not understand the proof in Lemma 3 Proof line 2 to 3. Why

$\text{Tr}\left(\phi\left(\mathbf{x}_t\right)\phi\left(\mathbf{x}_t\right)^\top\mathbf{W}^{\left(t+1\right)}\mathbf{W}^{\left(t+1\right)\top}\right)-\text{Tr}\left(\phi\left(\mathbf{x}_t\right)\phi\left(\mathbf{x}_t\right)^\top\mathbf{W}^{\left(t\right)}\mathbf{W}^{\left(t\right)\top}\right)$

less or equal than

$\text{Tr}\left(\phi\left(\mathbf{x}_t\right)\phi\left(\mathbf{x}_t\right)^\top\left(\mathbf{W}^{\left(t+1\right)}-\mathbf{W}^{\left(t\right)}\right)\left(\mathbf{W}^{\left(t+1\right)}-\mathbf{W}^{\left(t\right)}\right)^\top\right)$ ?

**Relation To Broader Scientific Literature:**

After Wołczyk, this work propose another kind of vision of the CRL, which will helpful for following works. I think this work tells us that the CRL is a very different question with the continual classification problem.

**Theoretical Claims:**

I am interested in understanding the extent to which Assumption 1 and Assumption 3 are sustained throughout the experiments. Specifically, I would like to inquire whether the configuration of $\Lambda$ in the experimental setup is sufficient to satisfy the conditions stipulated in Assumption 3.

---

> ### Author Rebuttal · Authors · 2025-04-01
>
> Thank you for your valuable review and questions. Below we respond to the comments and raised questions.
>
> ***Methods And Evaluation Criteria***
>
> > What does "task changes" in Algorithm A.2 mean and how does the agent know this?
>
> Admittedly, the presented algorithm still requires the task boundary information to reset the planner state. We save the planner state (memory, please see line 245) to mainly reuse the previous planning for better initialization, which is an implementation choice. We could also discard the memory and re-plan every time step, and therefore remove this condition.
>
> > Will the sparsity change after line 12-13 of Algorithm A.2?
>
> Yes, the sparsity of the matrix will change after the update. However, each update is guaranteed to be sparse because the sparsity of $\phi_s(x_t)$ is pre-determined, which also bounds the computation per update.
>
> ***Theoretical Claims***
> >whether the chosen $\lambda$ in the experiments is adequately configured to ensure that Assumptions 1 and 3.
>
> Both Assumption 1 and 3 depend on selecting a sufficiently large $\\lambda$ to control the growth of the feature-space covariance and the norms of sub-block inverses. Specifically, Assumption 1 ensures that the empirical average of feature outer products stays close to those from new data, a condition more easily met when $\\lambda$ is larger. Meanwhile, Assumption 3 employs $\\lambda$ in the regularized inverse $\\bigl(\\mathbf{A}\_{ss}\^{(t)} + \\tfrac{1}{\\lambda}\\mathbf{I}\\bigr)\^{-1}$ to keep $K$ within reasonable bounds. In practice, an appropriate $\\lambda$ helps maintain both assumptions throughout experiments by mitigating outlier effects in the feature space and ensuring that inverse sub-blocks remain well-conditioned. However, because the ideal value of $\\lambda$ can vary by environment or dataset, some tuning is typically required.
>
>
> ***Experimental Designs Or Analyses***
> > why the horizon $H$ is set to 15?
>
> This hyperparameter would trade off the planning accuracy and the computational requirement. Setting $H$ too small will get efficient planning but the planned action can be short-sighted; a longer $H$ could incur larger planning cost, but the policy can better optimize long-term rewards. However, since we plan with a learned world model, the model error can compound along the planning horizon, so there is no monotonic benefit from using longer horizons. $H=15$ is an empirical value which is obtained from our pilot trials.
>
>
> ***Questions For Authors***
>
> Thank you for pointing out the mistake. We missed an extra term in our proof. Below is the corrected derivation:
> \\begin{align}
>     {} &
>     \\mathrm{Tr}(\\phi(\\mathbf{x}\_t)\\phi(\\mathbf{x}\_t)\^\\top\\mathbf{W}\^{(t+1)}\\mathbf{W}\^{(t+1)\\top})
>     -
>     \\mathrm{Tr}(\\phi(\\mathbf{x}\_t)\\phi(\\mathbf{x}\_t)\^\\top\\mathbf{W}\^{(t)}\\mathbf{W}\^{(t)\\top})
>     \\\\
>     = {} &
>     \\mathrm{Tr}\\left(\\phi(\\mathbf{x}\_t)\\phi(\\mathbf{x}\_t)\^\\top\\left(\\mathbf{W}\^{(t+1)} - \\mathbf{W}\^{(t)}\\right)\\left(\\mathbf{W}\^{(t+1)} - \\mathbf{W}\^{(t)}\\right)\^\\top\\right) \\\\
>     & +
>     2\\mathrm{Tr}\\left(\\phi(\\mathbf{x}\_t)\\phi(\\mathbf{x}\_t)\^\\top\\left(\\mathbf{W}\^{(t+1)} - \\mathbf{W}\^{(t)}\\right)\\mathbf{W}\^{(t)\\top}\\right)
> \\end{align}
>
> Notice that by the definition of $\\Delta\_t$, we have
> \\begin{align}
>     {} &
>     2\\mathrm{Tr}\\left(\\phi(\\mathbf{x}\_t)\\phi(\\mathbf{x}\_t)\^\\top\\left(\\mathbf{W}\^{(t+1)} - \\mathbf{W}\^{(t)}\\right)\\mathbf{W}\^{(t)\\top}\\right)
>     \\\\
>     \\leq {} &
>     2\\mathrm{Tr}\\left(\\phi(\\mathbf{x}\_t)\\phi(\\mathbf{x}\_t)\^\\top\\Delta\_t\\mathbf{W}\^{(t)\\top}\\right)
>     \\\\
>     = {} &
>     2\\mathrm{Tr}\\left(\\underbrace{\\phi(\\mathbf{x}\_t)\\phi(\\mathbf{x}\_t)\^\\top\\left(\\sum\_{i=1}\^{t}\\phi(\\mathbf{x}\_i)\\phi(\\mathbf{x}\_i)\^\\top + \\frac{1}{\\lambda}\\mathbf{I}\\right)\^{-1}}\_{\\preceq \\mathbf{I}}\\phi(\\mathbf{x}\_t)\\mathbf{y}\_t\^\\top\\mathbf{W}\^{(t)\\top}\\right)
>     \\\\
>     \\leq {} &
>     2\\mathrm{Tr}\\left(\\phi(\\mathbf{x}\_t)\\mathbf{y}\_t\^\\top\\mathbf{W}\^{(t)\\top}\\right).
> \\end{align}
>
> So the correct bound should be $f\_t(\\mathbf{W}\^{(t+1)}) - f\_t(\\mathbf{W}\^{(t)}) \\leq \\mathrm{Tr}\\left(\\phi(\\mathbf{x}\_t)\\phi(\\mathbf{x}\_t)\^\\top\\Delta\_t\\Delta\_t\^\\top\\right) + \\frac{4}{t}\\mathrm{Tr}\\left(\\phi(\\mathbf{x}\_t)\\mathbf{y}\_t\^\\top\\mathbf{W}\^{(t)\\top}\\right)$, which differs from the original bound where the coefficient of the second term is $\\frac{2}{t}$.
> We will revise the proof in our manuscript.

---

> > ### Comment · Reviewer_WQD5 · 2025-04-02
> >
> > Thank you for your response. The authors' clarification has satisfactorily addressed my concerns. I have no further questions.

---

### Official Review · Reviewer_ULR9 · 2025-03-10

**Overall Recommendation:** 3

**Summary:**

The paper presents a Follow-The-Leader-based online world model, implemented as a composition of a learnable linear layer and a fixed-weight non-linear layer, that is used to solve the continual reinforcement learning problem as a part of model predictive control. The world model has a regret bound of $\mathcal{O}(\sqrt{K^2D\log(T)})$ under certain assumptions. The paper also introduces the $\textit{Continual Bench}$ benchmark and demonstrates the superiority of their online world model.

## update after rebuttal

I will maintain my score, given that the rebuttal has not significantly changed my opinion of the paper and my rebuttal comment is unaddressed.

**Claims And Evidence:**

The paper's main claims are supported as its method outperforms agents built on deep world models on their $\textit{Continual Bench}$ benchmark, and their regret proof is sound given their assumptions.

Section 2 claims that "image-based environments demand prohibitive computation resources, and the lack of meaningful overlapping prevents us from evaluating the transfer of resources". This claim isn't necessarily true since Atari environments have been successfully tackled, for offline learning, even back in 2015 with much less compute [1]. It's unclear why online learning should be more challenging w.r.t. computational resources.

A minor criticism is that the line 128 states that continual agent may not know the number of tasks, but the paper's algorithm in the appendix makes use of task boundaries to reset $\mu$.

[1] Mnih, V., Kavukcuoglu, K., Silver, D., Graves, A., Antonoglou, I., Wierstra, D., & Riedmiller, M. (2013). Playing atari with deep reinforcement learning. arXiv preprint arXiv:1312.5602.

**Essential References Not Discussed:**

To my knowledge, the paper covers all essential references.

**Experimental Designs Or Analyses:**

Yes. The evaluation of OA and baselines on the Continual Bench benchmark was insightful.

I have one concern, which is that the Continual Bench experiments only train the learning algorithms on 100 episodes per task. To my knowledge, 100 is not a lot of gradient steps for RL algorithms to maximally learn from an environment, and the success rate plot in Figure 5 shows that when task learning is over many models' performance curves don't plateau. While this is not a critical issue for continual learning since the setup does not assume the models can be trained to convergence, repeating the experiment for a larger episode count would yield valuable insights into how different world model performances change when given more chance to learn.

**Methods And Evaluation Criteria:**

Yes. However, the Continual Bench's number of tasks is only 6, which is considerably smaller than that of Continual World [1], which is 20. It would be ideal to test on a longer number of tasks, especially since the paper's learning algorithm OA can run out of capacity due to being a fixed-size linear model.

[1] Wołczyk, M., Zając, M., Pascanu, R., Kuciński, Ł., & Miłoś, P. (2021). Continual world: A robotic benchmark for continual reinforcement learning. Advances in Neural Information Processing Systems, 34, 28496-28510.

**Other Comments Or Suggestions:**

- Specify what $g, \delta$ are in Eq 8.

**Other Strengths And Weaknesses:**

- It is unclear how OA can scale to more complex problems, as it seems both OA and CEM are suited for simple, low-dimensional state and action spaces.

**Questions For Authors:**

- How applicable is their assumption 1 for theorem 1? Doesn't it assume the observations become more and more similar to past observations? In this case, wouldn't the constant regret assumption not apply to highly nonstationary environments?

- How effective is the sparse online world model learning update is for discrete state space? It seems to hinge on L2 loss for derivations to hold.

- How does continual bench change reward exactly? When learning a new task does the model not receive reward for previous tasks?

**Relation To Broader Scientific Literature:**

The paper adds to the array of recent work that investigates continual reinforcement learning through the lens of model stability and plasticity. To my knowledge, this is the first work that investigates online world model learning applied to continual reinforcement learning.

**Theoretical Claims:**

I have read the proof of Theorem 1 present in the appendix, but not extremely carefully.

---

> ### Author Rebuttal · Authors · 2025-04-01
>
> Thank you for your valuable review and questions. Below we respond to the comments and raised questions.
>
> ***Claims And Evidence***
> > The claim that "image-based environments...
>
> We agree that the Atari games were solved even back in 2015. However, it typically takes millions to billions of simulated time steps for learning a single task, which amounts to hours to days of wall-clock time even in a distributed training setup [1,2]. Besides, in the online continual RL setting, we could not parallelize the data collection since the online agent only has a single lifetime. This constraint, together with the fact that we need to learn multiple tasks sequentially in continual RL, could make the computation required by image-based experiments quite expensive, and therefore we opt for the state-based environments.
>
> > Line 128 states that continual agent...
>
> Admittedly, the presented algorithm still requires the task boundary information to reset the planner state. We save the planner state (memory, please see line 245) to mainly reuse the previous planning for better initialization, which is an implementation choice. We could also discard the memory and re-plan every time step, and therefore remove this condition.
>
>
> ***Methods And Evaluation Criteria***
> We agree that 6 tasks are limited. However, we hope to clarify that Continual World's CW20 is actually CW10 repeated twice. Although we only include 6 sequential tasks, we cover different objects like window, peg, button, door and faucet, which have similar difficulty level to CW10's tasks. As a future work, we plan to extend the Continual Bench to include more tasks, making it more useful for evaluating life-long learning online agents.
>
> ***Experimental Designs Or Analyses***
>
> We agree with your points. We set the maximum number of episodes per task as 100 episodes due to the limited computational budget and time constraint. We'd love to conduct experiments for longer episodes once the computational resources are available.
>
> ***Other Strengths And Weaknesses***
>
> Thanks for raising this important question. We admit that the existing algorithm has difficulty scaling up to high-dimensional problems, as we have discussed in Appendix D. Nevertheless, we hope the online world model learning + planning framework can be useful for developing online agents for more complex problems. We hope to continue this line of research to develop more capable model classes for learning world models online, and design more efficient approximate planning algorithms.
>
> ***Other Comments Or Suggestions***
>
> Sorry for missing this. $g$ is the goal state and $\delta$ is a small number to determine whether the current state reaches the goal (typically 0.005 in our experiments). We will add the explanation in our revision.
>
> ***Questions For Authors***
> > How applicable is their assumption 1 for theorem 1?...
>
> We'd like to clarify that the condition $\\sup\_{\\mathbf{x}}\\|\\phi(\\mathbf{x})\\phi(\\mathbf{x})\^{\\top} - \\frac{1}{t}\\sum\_{i=1}\^t\\phi(\\mathbf{x}\_i)\\phi(\\mathbf{x}\_i)\^{\\top}\\|\_2 \\leq \\frac{1}{\\lambda t}$ does not require that new observations must become more and more similar to features that have already been seen. Instead, it ensures that any single feature vector has negligible impact on the overall covariance once enough data is gathered, enabling robust and consistent estimates.
>
> We acknowledge that this analysis may be less suited to highly nonstationary environments. However, if the environment evolves smoothly enough that each new “epoch” of data has a bounded impact on the overall covariance after sufficient samples, the same theoretical guarantees should apply.
>
> > How effective is the sparse online world model learning update for discrete...
>
> We use L2 loss primarily for analytical simplicity. Similar guarantees hold with other losses (e.g., cross-entropy for discrete state space cases) by maintaining the same assumptions. Our L2-based derivation does not rely on the state continuity and is not a fundamental constraint. Even with discrete states, one can also embed them in a continuous feature space (e.g., one-hot or learned embeddings) and apply the same incremental updates.
>
> > How does continual bench change reward exactly?
>
> When the environment switches to a new task, the observation and the reward signal of the **new** task will be provided to the agent, and the agent will **not** receive reward for previous tasks. So the agent only has a single stream of experience, and it is expected to learn from it without forgetting how to solve old tasks.
>
> ---
>
> [1] Espeholt, Lasse, et al. "Impala: Scalable distributed deep-rl with importance weighted actor-learner architectures." International conference on machine learning. PMLR, 2018.
>
> [2] Kapturowski, Steven, et al. "Recurrent experience replay in distributed reinforcement learning." International conference on learning representations. 2018.

---

> > ### Comment · Reviewer_ULR9 · 2025-04-02
> >
> > Thank you for your clarifications. Out of curiosity, do you expect the results to change significantly if the agent continues to receive rewards for previous tasks? One could argue that is the training regime that many continual learning benchmarks operate under, even though they typically cannot revisit old tasks, at least in their training stream without experience replay.

---

### Official Review · Reviewer_7VEQ · 2025-03-12

**Overall Recommendation:** 4

**Summary:**

In this paper, the authors focus on the problem of Continual Reinforcement Learning, solving multiple tasks that are presented in sequence. In practice, this is a difficult problem because conventional methods often lead to Catastrophic Forgetting. To this end, the authors propose a model-based agent that learns via a Follow-The-Leader approach. The authors justify their approach through mathematical reasoning and proof. They then evaluate their approach on a multi-task setup (Continual Bench) based on Meta-World, demonstrating promising results over the state of the art.

## update after rebuttal
In light of the other reviewer's comments, I keep my assessment of accept.

**Claims And Evidence:**

Yes. The basic claims made in the paper seem to be justified empirically in the evaluations, at least in the specific dataset used (Continual Bench). The quantitative results are strong and show promise.

**Essential References Not Discussed:**

No. Not to my knowledge.

**Experimental Designs Or Analyses:**

Yes. The experimental design is sound.

**Methods And Evaluation Criteria:**

Yes, albeit only one dataset was used (Continual Bench), and this dataset was created by the authors. If there were a way to try other Continual Learning benchmarks, that would strengthen the case of the paper.

**Other Comments Or Suggestions:**

None.

**Other Strengths And Weaknesses:**

The authors focus on an important problem in Reinforcement Learning (Multi-task/Continuous Learning) and propose an approach with strong theoretical justification. Furthermore, they present promising results and a new dataset for future research. I believe this paper would be a good contribution to the conference. The paper's argument could be strengthened with evaluation on more datasets/domains; although it may be difficult to find appropriate continual learning domains as the authors note.

**Questions For Authors:**

None.

**Relation To Broader Scientific Literature:**

Compared to prior work, the authors propose a new method for Continual Reinforcement Learning in addition to a new dataset (Continual Bench)

**Theoretical Claims:**

The theoretical claims made in the paper appear to be correct, although it's possible I may have missed details.

---

> ### Author Rebuttal · Authors · 2025-04-01
>
> Thank you for your supportive review, as well as the suggestions that we should evaluate the method on additional datasets/benchmarks. The main challenge of doing so is that there is no appropriate continual RL test suite to our knowledge. This is also the motivation for us to discuss the importance of unified world dynamics and to propose a new benchmark (Continual Bench).
>
> Nevertheless, we will continue working on this area and test on newly developed benchmarks if any. Thank you again for your feedback!

---

### Official Review · Reviewer_mVEi · 2025-03-14

**Overall Recommendation:** 4

**Summary:**

Update after rebuttal:

I have read the author responses and reviews. I would like to maintain my score recommending acceptance.


Summary:

The paper proposes an approach to Continual Reinforcement Learning (CRL) through the development of an Online Agent (OA) that leverages online world models. The central idea is to address catastrophic forgetting—a major challenge in CRL—by employing Follow-The-Leader (FTL) shallow models to capture world dynamics, paired with model predictive control (MPC) for planning. The authors introduce Continual Bench, a new benchmark designed to test both forgetting and transfer in CRL environments. Empirical results show that OA outperforms strong baselines built on deep world models with traditional continual learning techniques.


Overall impression:

I really like the way the paper is written. It is quite modular and clear. Sufficient literature has been cited throughout the text. The core idea of learning a unified world dynamics that can be shared among different tasks is appealing for transfer. I think the community will benefit from this paper.

**Claims And Evidence:**

This paper claims that the OA agent can solve CRL tasks incrementally without catastrophic forgetting by planning with online world models. OA surpasses deep world models combined with various continual learning methods. Regret bounds for the sparse online model learning process are formally derived and justified with key assumptions (like feature mappings stabilization and bounded inputs/outputs).

**Essential References Not Discussed:**

None

**Experimental Designs Or Analyses:**

The setup allows studying both forgetting (by testing how agents retain old skills) and transfer (by evaluating knowledge reuse across similar tasks). Tasks are sequenced to maximize distributional shifts, testing how well models generalize without task-specific identifiers.

**Methods And Evaluation Criteria:**

The evaluation metrics used (average performance, regret, learning curves) seem appropriate. I really like the way the authors evaluate on all previously seen tasks even when learning on a particular new task to track catastrophic forgetting.

**Other Comments Or Suggestions:**

Typo:
Line 292: stat -> state

**Other Strengths And Weaknesses:**

Covered already.

**Questions For Authors:**

What is meaningful overlapping referred to? overlapping of what?

**Relation To Broader Scientific Literature:**

The work builds on model-based RL foundations (Sutton, 1990) and extends them into CRL.
It directly engages with recent FTL-based approaches (Liu et al., 2024) and compares against standard CL methods like EWC, SI, and PackNet. The benchmark Continual Bench critiques and refines prior environments like Continual-World (Wołczyk et al., 2021), addressing their limitations by ensuring consistent state spaces for testing transfer and forgetting.

**Theoretical Claims:**

The paper provides a regret bound for the sparse online model learning process, ensuring that OA's world model updates incrementally without forgetting past tasks.

---

> ### Author Rebuttal · Authors · 2025-04-01
>
> Thank you for your supportive review and questions. Below we respond to the comments and raised questions.
>
> ---
>
> ***Q1: What is meaningful overlapping referred to?***
>
> We intended to mean the overlapping of task attributes. For example, two Atari games may lack meaningful overlapping because they have distinct appearance and different underlying game logic. The lack of task attribute overlapping makes the study of transfer difficult. In our proposed environment (Continual Bench), we carefully designed a sequence of tasks that share certain attributes (e.g., moving the robotic arm, gripping objects), ensuring meaningful overlapping of task attributes.
>
> Thank you again for your question, and we will make the term clearer in our next revision, as well as fixing the typo you mentioned.

---

### Decision · Program_Chairs · 2025-05-01

**Decision:**

Accept (spotlight poster)

**Comment:**

This paper proposes an Online Agent (OA) for Continual Reinforcement Learning (CRL), leveraging Follow-The-Leader (FTL) world models and Model Predictive Control (MPC) to mitigate catastrophic forgetting. It also introduces Continual Bench, a new benchmark designed to evaluate both forgetting and transfer across tasks.

All reviewers agree that the paper tackles an important problem in CRL and is grounded in strong theoretical foundations. The OA approach is clearly articulated, with a regret bound derived under assumptions generally considered reasonable. The proposed Continual Bench is a valuable contribution with improved consistency over prior environments by maintaining a fixed state space. Empirical results are promising and support the proposed methodology.

There are some concerns. The benchmark includes only six tasks, all defined by the authors, which is relatively small compared to prior work such as *Continual World* (Reviewers 7VEQ and ULR9). This raises questions about the scalability of the approach. Additionally, OA’s reliance on shallow models may limit performance in more complex or high-dimensional settings (Reviewer ULR9). Finally, some assumptions made in the theoretical analysis could be better reflected or motivated in the experimental setup (Reviewer WQD5).

Overall, this paper presents a novel, theoretically sound approach to CRL, supported by a meaningful empirical evaluation. Despite some concerns about scalability and generalization, the contribution is considered valuable and relevant to the ICML community.